# EFFICIENT APPROXIMATE POSTERIOR SAMPLING WITH ANNEALED LANGEVIN MONTE CARLO

**Advait Parulekar**[*]
University of Texas at Austin

**Litu Rout**
University of Texas at Austin

**Karthikeyan Shanmugam**
Google DeepMind

**Sanjay Shakkottai**
University of Texas at Austin

## ABSTRACT

We study the problem of posterior sampling in the context of score based generative models. We have a trained score network for a prior $p(x)$, a measurement model $p(y|x)$, and are tasked with sampling from the posterior $p(x|y)$. Prior work has shown this to be intractable in KL (in the worst case) under well-accepted computational hardness assumptions. Despite this, popular algorithms for tasks such as image super-resolution, stylization, and reconstruction enjoy empirical success. Rather than establishing distributional assumptions or restricted settings under which exact posterior sampling is tractable, we view this as a more general "tilting" problem of biasing a distribution towards a measurement. Under minimal assumptions, we show that one can tractably sample from a distribution that is *simultaneously* close to the posterior of a *noised prior* in KL divergence and the true posterior in Fisher divergence. Intuitively, this combination ensures that the resulting sample is consistent with both the measurement and the prior. To the best of our knowledge these are the first formal results for (approximate) posterior sampling in polynomial time.

## 1 INTRODUCTION

Score-based generative models (Song & Ermon, 2020) including DALL-E (Ramesh et al., 2021), Stable Diffusion (Rombach et al., 2022), Imagen (Saharia et al., 2022), and Flux (Black Forest Labs, 2024), provide a powerful framework for sampling from complex data distributions. Given access to samples from a target distribution, these models learn a family of *smoothed score functions*, i.e., vector fields that estimate the gradient of the log-density of the data corrupted with varying levels of noise. Intuitively, these score functions can be used to map an image corrupted with a certain amount of noise to an image with less noise. Once such a family of score functions is learned, it can be used to iteratively denoise an image starting from pure noise and generate a sample from the data distribution.

The success of score-based generative models in capturing complex prior distributions has led to their widespread adoption in downstream tasks such as inpainting (Lugmayr et al., 2022), super-resolution (Kawar et al., 2022; Chung et al., 2022; Song et al., 2023; Rout et al., 2023; 2024), MRI reconstruction (Song et al., 2022b), and stylization (Hertz et al., 2024; Rout et al., 2025b;a). In these tasks, we begin with a prior $p$ specified to us through a large number of samples. We also have a likelihood or a reward model denoted by $R_y$ that indicates our preference at inference time, which is typically parameterized by a measurement $y$. The tasks is to obtain a sample from $p$ that is consistent with $R_y$.

In many practical scenarios, such as those mentioned above, the measurement model is given by $y = \mathcal{A}(x) + \eta$, where $\mathcal{A}$ is a known measurement operator and $\eta$ is noise. We seek a sample $x$ from the prior such that $y \approx \mathcal{A}(x)$. This is often implemented by using $R_y = \|\mathcal{A}(x) - y\|^2$ as a potential function and considering a KL penalty. Formally, this is equivalent to sampling from the

---

[*]advaitp@utexas.edu

tilted distribution $\mu_0$, which is defined as follows:

$$\mu_0 = \arg\min_\nu \mathbb{E}_\nu[R_y(X)] + \mathsf{KL}(\nu\|p) \implies \mu_0 \propto pe^{-R_y} \qquad \text{(Posterior Sampling)}$$

This paper explores the extent to which score networks trained to model the prior $p$ can be used for sampling the tilted distribution. We refer to this type of tilting as Posterior Sampling. Indeed, if $p$ is the prior, and $e^{-R_y}$ is a likelihood, then $pe^{-R}/Z$ is the posterior given the measurement $y$. This setting differs from traditional *conditional generation*, where conditioning variables (e.g., measurements) are fed as input to the score network. In contrast, our focus is on a *training-free* setup: given a measurement $y$ at inference time, we aim to sample from $p(x|y)$ using only a score network trained on the unconditional prior $p(x)$. While such networks are known to enable efficient sampling from $p(x)$ (Chen et al., 2023), our goal in this paper is to understand their role in sampling from $p(x|y)$.

There has been growing interest in establishing provable guarantees for posterior sampling. In general, we cannot directly use the score based generative models, because we cannot efficiently compute the posterior smoothed scores from the prior smoothed scores. While empirically successful methods often perform well in practice and implicitly aim to solve the posterior sampling problem, provable polynomial-time guarantees remain elusive. In fact, many of the efficient algorithms proposed (Chung et al., 2022; Rout et al., 2023) can be proven to be biased. A formal counterpoint was presented in Gupta et al. (2024), which showed that one could set up a posterior sampling problem to invert a (hypothesized) cryptographic one-way function, establishing cryptographic hardness. Intuitively, this hardness stems from the fact that posterior sampling is a composite sampling problem that encourages consistency with both a prior distribution as well as the measurement likelihood, which is difficult when the regions of highest likelihood have a small probability under the prior.

In light of this, recent work has focused on identifying sufficient conditions under which provable or asymptotically correct posterior sampling is possible, while avoiding such lower bounds (Bruna & Han, 2024; Xu & Chi, 2024). Instead, we take the view that exact posterior sampling might be a more difficult goal than we really need to achieve. In what sense can we tractably bias a sample from a prior towards a likelihood?

**Contributions.** We introduce a notion of posterior sampling that is possible in polynomial time, bypassing the hardness of sampling in $\mathsf{KL}$. We develop guarantees with our method *Annealed Langevin Monte Carlo* (ALMC, Algorithm 1) in the general regime where the influences of the prior and the likelihood might be in conflict. We start with a sample that disregards the prior entirely – emphasizing only consistency with the likelihood. This sample is then annealed towards the true posterior by drawing its marginal closer to the posteriors of progressively denoised priors. Other than at polynomially low noise levels, we show that using ALMC we can efficiently transition from the posterior of a noised prior to a posterior of a slightly less noised prior. This efficiency is captured by bounds on how quickly these posteriors can change as we vary the level of noising on the priors (Lemmas B.2, C.6), as well as regularity conditions on the posteriors themselves, being as they were posteriors on priors that are regularized by annealing (Lemmas C.5, C.7). This brings us to the two main contributions of our work,

   a. We show that an early-stopped Annealed Langevin Monte Carlo (ALMC) algorithm can track the posterior of a slightly noised prior in polynomial time in $\mathsf{KL}$, and thus sample from a distribution close to the *posterior for a noisy prior*.

   b. Although tracking the above path in $\mathsf{KL}$ beyond this point is generally intractable, we show that this early stopped distribution also has a low Fisher Divergence relative to the *true posterior*.

Our results require minimal assumptions (Assumptions 4.1) – that the prior should have Lipschitz score, be sub-Gaussian, and that the measurement operator $R_y$ should be smooth and convex. Our motivation for this pair of results stems from the phenomenon of "mode collapse", shown in the context of "unannealed" Langevin Monte Carlo for convergence in $\mathsf{FI}$ (Balasubramanian et al., 2022). Indeed, we show in Sections 2.1 and 4.1 that for a multimodal distribution (for example, a mixture of Gaussians), Fisher Divergence alone suffices only to guarantee a type of *local* convergence, and cannot generally provide any guarantees on the corresponding mode weights (e.g., mixture weights). Our early stopped $\mathsf{KL}$ guarantee for the posterior of a noised prior provides a notion of global correctness in density. Specifically, in the mixture-of-Gaussian setting, we show that we can

explicitly avoid mode collapse (Section 4.1). Taken together, these results provide a response to the intractability of posterior sampling in KL.

**Notation:** We use $p_0$ to denote a prior, $R_y$ (or $R$) to denote a likelihood, and $\mu_0 \propto p_0 e^{-R}$ to denote a posterior. We use $\gamma$ to refer to a standard Gaussian. For time $t$, $p_t$ denotes the Gaussian smoothed prior (or noised prior) with density $p_t(x) = e^{td} p_0(e^t x) * \gamma$, where $d$ is the ambient dimension ($x \in \mathbb{R}^d$), and $*$ is the convolution operator. Similarly, we define $(\mu_0)_t(x) = e^{td} \mu_0(e^t x) * \gamma$ (the noised true posterior) and $\mu_t \propto p_t e^{-R}$ (the posterior of the noised prior). We have $\mathsf{KL}(\alpha \| \beta) = \mathbb{E}_\alpha [\log \alpha/\beta]$, $\mathsf{TV}(\alpha, \beta) = \sup (\alpha(A) - \beta(A))$ where the supremum is over all measurable sets $A$.

## 1.1 RELATED WORKS

**Sampling:** We refer the reader to Chewi (2023) for an exposition of works on sampling. There are strong connections between sampling and optimization, explored in various places including Wibisono (2018). Approximately, we can think of Langevin Monte Carlo (LMC) for sampling as corresponding to Gradient Descent for optimization, and log-concave distribution correspond to convex functions. More recently, denoising diffusion models (Ho et al., 2020; Song et al., 2022a; Song & Ermon, 2020; Song et al., 2021) begin with a noisy image and iteratively denoise to get a sample. This is efficient, but requires a trained *score network*. Finally, the idea of running LMC towards a changing target distribution is related to works on annealing and tempering (Marinari & Parisi, 1992; Hajek & Sasaki, 1989). One can think of DDPM (Ho et al., 2020) as doing this using "heat" in a different way - by Gaussian convolution of the measures (adding heat to the particles).

**Posterior Sampling:** This is a very active area of research, with a number of different approaches. Some methods try to estimate the posterior score $\nabla \log p_t(x_t|y)$ directly (Chung et al., 2022; Rout et al., 2024; Song et al., 2022b); we refer the reader to Daras et al. (2024) for a more extensive treatment. The barrier for provable results with these methods is that getting the scores for the noisy posteriors exactly can be computationally intractable. Others use a sequence of operations alternatingly aligning the iterate with the measurement and prior (Cordero-Encinar et al., 2025; Xu & Chi, 2024; Wu et al., 2024; Rout et al., 2025b). These are variants of "Split-Gibbs" sampling, which has a biased stationary distribution to which there are generally asymptotic convergence results, but no finite time, or even unbiased, guarantees. An exception is Wu et al. (2024), which gets an "average" Fisher Divergence guarantee. There are also particle filtering methods (Chung et al., 2022; Dou & Song, 2024), which use Sequential Monte Carlo to estimate the posterior using a set of particles. Here the guarantees are in the limit as the number of particles grows to infinity. Indeed, formal guarantees appeared to be elusive, and a result of Gupta et al. (2024) showed that posterior sampling is intractible in the worse case under the existence of a one way function. More recently Bruna & Han (2024) showed that posterior sampling can also be reduced to sampling from an ill-conditioned ising model, which is known to be impossible unless $\mathsf{NP} = \mathsf{RP}$.

**Fisher Divergence bounds:** In the classical (that is, without a trained score network) sampling literature, recently Balasubramanian et al. (2022); Wibisono (2025) proposed using Fisher Divergence to capture the phenomenon of metastability, which can be thought of as a type of approximate first order convergence.

## 2 BACKGROUND

**Gradient Flows:** Consider a Markov process $X_t$ described by the SDE below. Let $\rho_t$ denote the law of $X_t$, and let $B_t$ denote a Wiener process. The measure $\rho_t$ can be thought of as evolving according to a vector field $v_t$. This flow can be expressed using the Fokker-Planck equation as shown to the right below.

$$dX_t = v_t(X_t)\, dt + \sqrt{2} dB_t \iff \partial_t \rho_t = -\nabla \cdot (\rho_t v_t) + \Delta \rho_t \qquad \text{(Fokker-Planck)}$$

An absolutely continuous path $t \mapsto \rho_t$ is *generated* by $v_t$ if the Fokker-Planck equation is satisfied. Also, for any absolutely continuous path, there is a canonical "minimal" velocity field that generates it. We refer the reader to Ambrosio & Savaré (2007) for a detailed exposition.

**Langevin Dynamics:** Langevin Dynamics refers to the SDE

$$dX_t = \nabla \log \pi(X_t)\, dt + \sqrt{2} dB_t \iff \partial_t \rho_t = \nabla \cdot (\rho_t \nabla \log \frac{\rho_t}{\pi}) \qquad \text{(Langevin)}$$

It was noted in Jordan et al. (1998) that the law of the process is a gradient flow for the KL divergence functional $\mathsf{KL}(\cdot\|\pi)$ in the space of probability measures endowed with a Wasserstein metric. Convergence of $\rho_t$ to $\pi$ is characterized by a log-Sobolev inequality (LSI). Let FI denote the Fisher divergence (defined below), then the LSI states

$$\forall\, \rho,\ \mathsf{KL}(\rho\|\pi) \le \frac{1}{\alpha_\pi}\,\mathsf{FI}(\rho\|\pi) \qquad \mathsf{FI}(\rho\|\pi) = \mathbb{E}_\rho\|\nabla \log \frac{\rho}{\pi}\|^2 \qquad (\alpha_\pi\text{-LSI})$$

While log-Sobolev inequalities are usually difficult to establish tightly, one can show that a measure whose negative log-density is $\frac{1}{\alpha_\pi}$-strongly convex satisfies $\alpha_\pi$-LSI (Bakry et al., 2014). If a measure $\pi$ satisfies a log-Sobolev inequality, one can show that Langevin Dynamics enjoys linear convergence in KL (Vempala & Wibisono, 2022), specifically that

$$\mathsf{KL}(\rho_t\|\pi) \le e^{-2\alpha_\pi t}\mathsf{KL}(\rho_0\|\pi)$$

However, even for "simple" distributions like a mixture of two well-separated Gaussians, the LSI could have a very bad constant (in this case, exponentially small in the separation; see for instance Remark 3 in Chen et al. (2021)). This often prohibits the use of Langevin Monte Carlo.

**Reversing the Flow:** Modern score based generative models sample from a prior distribution $\pi$ by training a neural network to learn the flow that would *reverse* the forward Gaussian Langevin flow. Langevin Dynamics for a Gaussian is also called the Ornstein–Uhlenbeck (OU) process

$$dX_t = -X_t dt + \sqrt{2}dB_t \iff \partial_t \rho_t = \nabla \cdot (\rho_t(\nabla \log \rho_t + x)) \qquad \text{(OU)}$$

Sampling $X_0 \sim \pi_0$ and running the above SDE for time $t$ results in $X_t \sim \pi_t$. We note that $\pi_t$ can explicitly be written as: $\pi_t(x) = e^{td}\pi_0(e^t x) * \gamma$. From classical literature on reversing SDEs (Anderson, 1982), we know the following:

$$\underbrace{dX_t = -X_t dt + \sqrt{2}dB_t}_{\text{forward process}} \iff \underbrace{dX_t^{\leftarrow} = (X_t^{\leftarrow} + 2\nabla \log \pi_t(X_t^{\leftarrow}))\,dt + \sqrt{2}dB_t}_{\text{reverse process}}. \qquad (1)$$

One can begin at $X_0^{\leftarrow} \sim \pi_T$ and run the reverse process to get $X_t^{\leftarrow} \sim \pi_{T-t}$ until $X_T^{\leftarrow} \sim \pi_0$. In fact, the random variables $X_t$ and $X_{T-t}^{\leftarrow}$ have the same law. The key to being able to implement this process is the use of the *score* $\nabla \log \pi_t$. Due to Tweedie's lemma (Robbins, 1956):

$$\sqrt{1-e^{-2t}}\,\nabla \log \pi_t(x) = e^{-t}x_t - \mathbb{E}\left[x|e^{-t}x + \sqrt{1-e^{-2t}}\eta = x_t\right] \qquad \eta \sim \gamma \quad \text{(Tweedie)}$$

These can be learned using a simple variational characterization of least squares regression. Consider a family of models $s_\theta(x,t)$ parameterized by $\theta$. We find

$$\theta^* = \arg\min \mathbb{E}_{x,\eta}\|x - s_\theta(x + \sigma_t \eta, t)\|^2 \qquad (2)$$

From here, we can estimate the score $\nabla \log \pi_t(x)$ as $\nabla \log \pi_t(x) \approx \frac{s_{\theta^*}(x,t)-x}{\sigma_t^2}$ [1].

**Annealed Langevin:** Rather than using the reverse process specified above, one could use an *"annealed"* Langevin Dynamics. Unlike traditional Langevin where the drift of the SDE is given by the score of a single density, here the density evolves over time as follows:

$$dX_t = \nabla \log \pi_t(X_t)dt + \sqrt{2}dB_t \qquad \text{(Annealed Langevin)}$$

Unlike the true reverse SDE, this annealed Langevin incurs a bias that stems from the fact that it never quite reaches $\pi_t$ by time $t$. The bias is characterized in Guo et al. (2024), Cordero-Encinar et al. (2025), where it is shown to be related to the *action* of the path $\pi_t$ through the space of distributions. Specifically, for the path $\pi_t$ described above, the action is bounded in Cordero-Encinar et al. (2025) by a quantity that is independent of any functional inequalities.

In fact[2], any path $t \mapsto \pi^t$ with velocity field $v_t$ can be efficiently sampled from by starting with $X_0 \sim \pi^0$ and running $\dot{X}_t = v_t(X_t) \implies X_t \sim \pi^t$. However, for an arbitrary path $t \mapsto \pi^t$, it may not be easy to initialize $X_0 \sim \pi^0$, or to compute the corresponding velocity field $v_t$. Implementing the ODE also incurs a discretization bias.

---

[1] There is a line of work analyzing the propagation of score matching errors into the sampling distribution (Chen et al., 2023; Lee et al., 2023). Because of our interest in the posterior sampling problem, we will assume that we have the exact prior score.

[2] We use a superscript here to emphasize that $\pi^t$ need not be the marginal of an OU process, like $\pi_t$.

**Remark 2.1** (Action). *We can think of the action of a path as giving the run time of sampling along it using annealed Langevin. Different paths connecting $\pi^0$ and $\pi^T$ coming from different fields $v_t$ give different actions. Some $v_t$ lead to paths that are fast but difficult to compute, like the optimal transport path, or the constant speed geodesic connecting $\pi^0$ to $\pi^T$. This path can be shown to have the least action over all paths, but to implement this we would need to compute the optimal transport map. On the other hand, Annealed Langevin has a large action but could be easier to implement.*

**Discretization:** Langevin Monte Carlo is an efficient discretization of Langevin Dynamics, where the drift is fixed over small intervals of time. Suppose we run our algorithm for time $T$, and suppose our discretization step size is $\delta$. Let $B_t$ denote a Wiener Process. We have the following "interpolated" process

$$dX_t = \nabla \log \pi(X_{k\delta}) \, dt + \sqrt{2} \, dB_t, \qquad t \in [k\delta, (k+1)\delta)$$

We can integrate this between $k\delta$ and $(k+1)\delta$ to get

$$X_{(k+1)\delta} = X_{k\delta} + \delta \nabla \log \pi(X_{k\delta}) + \sqrt{2}(B_{(k+1)\delta} - B_{k\delta}) \qquad \text{(LMC)}$$

We refer to this as running LMC *towards* $\pi$. Similarly, Annealed Langevin has the corresponding interpolation $dX_t = \nabla \log \pi_k(X_{k\delta}) \, dt + \sqrt{2} \, dB_t$ for $t \in [k\delta, (k+1)\delta)$, which can be discretized as

$$X_{(k+1)\delta} = X_{k\delta} + \nabla \log \pi_{k\delta}(X_{k\delta})\delta + \sqrt{2\delta} \, (B_{(k+1)\delta} - B_{k\delta}) \qquad \text{(Annealed LMC)}$$

**Remark 2.2** (Annealing). *There are two notions of annealing in the context of sampling. The first is temperature annealing, where the diffusive term of the SDE (Langevin) is modified to be $\sqrt{2/\log(2+t)}$ (Geman & Hwang, 1986). Second is Gaussian annealing, where the diffusive term is fixed, but the drift term of (Langevin) is modified by using the score of a smoothed prior. Indeed, the continuous time variant of DDPM (Song & Ermon, 2020) is such an annealing and Algorithm 1 is an archetype of the latter type of annealing for posterior sampling.*

## 2.1 Local Mixing and Metastability

Recall the interpretation of Langevin Dynamics as gradient flow in the space of measures towards a minimum of the functional $\mathsf{KL}(\rho\|\pi)$. There is only one global minima corresponding to the correct distribution: $\mathsf{KL}(\rho\|\pi) = 0 \implies \rho = \pi$. If we view the relative Fisher information $\mathsf{FI}(\rho, \pi)$ as a gradient norm in this analogy, one can ask whether we can quickly find a first order *approximately* stationary point $\rho$ satisfying $\mathsf{FI}(\rho, \pi) < \epsilon$. It is shown in Balasubramanian et al. (2022) that LD achieves $\mathsf{FI}(\overline{\rho}_t, \pi) < \epsilon$ in polynomial time $\mathcal{O}(d^2/\epsilon^2)$ for the *average* iterate, that is $\overline{\rho} = \frac{1}{T}\int \rho_t dt$. We remark that this convergence is independent of $\mathsf{LSI}$, but describes a weaker type of convergence as discussed below.

There is a sense in which $\mathsf{FI}$ convergence ensures local mixing within "modes" of a distribution. Take two distributions $\gamma_1, \gamma_2$. Let $\gamma_{1|\mathcal{B}_\varepsilon(x)}$ (respectively, $\gamma_{2|\mathcal{B}_\varepsilon(x)}$) denote the distribution $\gamma_1$ conditioned on being within a ball of radius $\varepsilon$ around the point $x$. In Lemma D.1, we show that for small enough $\varepsilon$:

$$\mathbb{E}_{X\sim\gamma_1}\mathsf{KL}\big(\gamma_{1|\mathcal{B}_\varepsilon(X)}\|\gamma_{2|\mathcal{B}_\varepsilon(X)}\big) \lesssim \varepsilon\mathsf{FI}(\gamma_1, \gamma_2) \qquad \text{(Pointwise LSI)}$$

In other words, *conditioned on being within a small radius of any point*, the two distributions match *in KL*, on average[3]. In a distribution with multiple separated modes, this means that conditioned on any specific mode, the sampler is accurate, even in $\mathsf{KL}$. For intuition, consider a distribution that has multiple modes (e.g., a mixture of Gaussians). The $\mathsf{FI}$ convergence implies that if initialized close to one of the modes, LMC will converge quickly to a sample "from this mode". Notably, however, in this setting, $\mathsf{FI}$ convergence is not very sensitive to the *weights* of the modes because the $\mathsf{FI}$ involves a gradient operation on the log-density, which makes it insensitive to mode weights. Thus, this is too weak to ensure a global convergence. We further discuss this in Remark 4.1 in the context of posterior sampling.

---

[3]If the standard $\mathsf{LSI}$: $\mathsf{KL}(\gamma_1\|\gamma_2) \lesssim \mathsf{FI}(\gamma_1, \gamma_2)$ were to hold, that would be the "global" analog of this result. However, the setting of regions of low density in between high density regions that is typical of multimodal distributions precludes such an LSI.

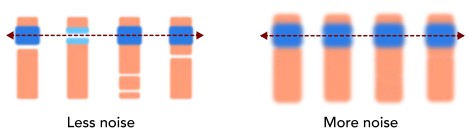

Figure 1: Hardness of posterior sampling: In this instance, the prior is represented by the orange region, we measure a coordinate specified by the red arrow. The posterior is represented by the blue region.

Less noise          More noise

## 2.2 POSTERIOR SAMPLING

The discussion thus far has been about the classical sampling problem – we want to sample from $\pi$ given $\nabla \log \pi$ or $\nabla \log \pi_t$. In the posterior sampling problem, we also have a likelihood $R$, and we would like to sample from $\mu = \pi e^{-R} / \int \pi e^{-R}$. There is no immediate way to use the prior smoothed scores to get the posterior smoothed scores. Many approaches to posterior sampling (Section 3 of Daras et al. (2024)) proceed by trying to estimate $\nabla \log(\mu_0)_t$, but none establish a formal guarantee.

In fact, the *hardness* of sampling from a posterior has been established in recent works. Gupta et al. (2024) describes an instance in which sampling from the prior is tractable yet sampling from a posterior derived from a noisy linear measurement is intractable under a cryptographic hardness assumption. Bruna & Han (2024) reduces the posterior sampling problem to an Ising model in which the prior is a uniform distribution of the hypercube and shows hardness under standard computational hardness results. We will discuss this difficulty intuitively using the Figure 1.

Consider the following instance. The prior consists of a number of modes (in Figure 1, there are four, one corresponding to each of the vertical "bars"). The measurement is the vertical coordinate (one such measurement is represented by the red dotted line). In our case, the leftmost bar and the two to the right are consistent with the measurement, while the second from the left is not. However, we cannot use the scores $\nabla \log \pi_t$ from high noise levels $t$ to tell whether a specific mode is consistent. That is, high noise levels scores cannot distinguish between the true prior and a prior with a different pattern of consistency, say one in which every mode is consistent. For distinguishing this, only the low noise level scores are useful, but usually by the time we are using the low noise level scores in an algorithm, we have already committed to a mode and cannot drift our samples to other modes.

This suggests that we look at posterior sampling at two scales. At a local scale, the low noise level prior scores $\nabla \log \pi_t$ (combined with the gradients of the log-likelihoods $\nabla R(x)$) contain enough information to sample correctly conditioned on any small neighborhood, and the locality of such a task ensures that this can be achieved by an SDE in polynomial time. The difficulty with sampling truly in KL is that these local guarantees cannot be accurately stitched together. We will see that the high noise level scores can be used to "warm-start" the local sampling described above.

## 3 ANNEALED LANGEVIN MONTE CARLO FOR POSTERIOR SAMPLING

---

**Algorithm 1** Annealed Langevin Monte Carlo

---

**Input:** $x_T \sim \gamma$, rate $1/\kappa$, Warm Up period $T$, Warm Start period $T_{ws}$, step size $\delta$
**Output:** $x_0$
1: ▷ Warm Start, sample $X_T \sim \mu_T \approx \mu_\infty$
2: **for** $i = 1$ to $T$ **do**
3:     Sample $\eta_i \sim \gamma$
4:     $z_i = z_{i-1} - \delta(z_{i-1} + \nabla R(z_{i-1})) + \sqrt{2\delta}\, \eta_i$
5: **end for**
6: ▷ Annealing phase, track distributions $\{\mu_t\}$ from $T_{ws} \to 0$
7: $x_{T_{ws}\kappa/\delta} = z_T$
8: **for** $i = T_{ws}\kappa/\delta$ to $0$ **do**
9:     Sample $\eta_i \sim \gamma$
10:     $x_{i-1} = x_i + \delta(\nabla \log p_{\frac{i\delta}{\kappa}}(x_i) - \nabla R(x_i)) + \sqrt{2\delta}\, \eta_i$
11: **end for**

---

We construct a path $t \mapsto \mu_t$ of posteriors, with $\mu_t \propto p_t e^{-R}$ (that is, posteriors of noised priors). In Figure 3, this curve is represented by the blue curve between $\mu_{T_{ws}}$ and $\mu_0$. This path is absolutely

continuous (see Lemma B.2) and thus generated by some velocity field $v_t$. However, because we do not know $v_t$, we cannot use this field to traverse the curve. Our results bound the *action* of this path to show that Annealed LMC tracks a discretization of this continuous path. We denote a sample at time $t$ by $X_t$, and the associated distribution by $\rho_t$. There are two phases to our algorithm, as below.

**Warm Start:** We sample our initial point $X_0$ from a standard Gaussian $\gamma$, and run LMC for target $\gamma e^{-R}/Z$ for $\log \frac{1}{\epsilon}$ iterates. Because $R$ is convex, $\gamma e^{-R}$ is strongly log-concave, so efficient convergence to within $\epsilon$ in KL follows from prior work (Vempala & Wibisono, 2022). We can think of this warm start as biasing our samples towards the measurement. At this point, we have not aligned our samples at all with the prior.

**Annealing:** Starting from $\mu_{T_{ws}}$ with $T_{ws} \asymp \frac{1}{\epsilon^2} \log \frac{1}{\epsilon}$, we run Annealed LMC to track the distributions $\mu_t$ from $T_{ws}$ to 0. We use a parameter $\kappa$ to control the rate at which we move along this path. Moving slowly results in better agreement between the law of the iterate and the corresponding target.

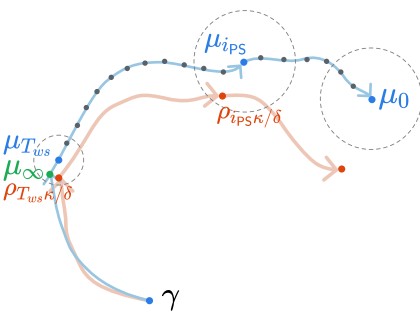

Figure 2: Beginning at $\gamma$, we use LMC to sample an initialization close to $\mu_\infty$. We then run the Annealed LMC tracking $\mu_t$. The blue path represents the target distributions, first the Langevin path from $\gamma \to \mu_\infty$, followed by $\{\mu_t\}$ from $\mu_\infty$ to $\mu_0$ (the true posterior). The orange curve indicates the laws of the iterates of LMC towards $\mu_\infty$ in the first phase, and the laws of the iterates of Annealed LMC towards $\{\mu_t\}$ for the second phase.

**A note on the rate $\kappa$:** From Lemma 4.3 we can sample from close to $\mu_{T_{ws}}$ in KL for $T_{ws} \asymp \frac{1}{\epsilon^2} \log \frac{1}{\epsilon}$ using LMC for target $\mu_\infty$. Rather than running the annealing backward at the same rate as the forward OU process, we slow it down[4] by a factor of $\kappa$. Our iterates go from $X_{T_{ws}\kappa/\delta} \to X_0$, the annealing targets go from $\mu_{T_{ws}} \to \mu_0$ in the continuous process, but in the discretized algorithm, the iterate $X_{i-1}$ uses target $\mu_{i\delta/\kappa}$. The law of the iterates $\rho_i$ goes from $\rho_{T_{ws}\kappa/\delta}$ to $\rho_0$.

**The pathology of $t \mapsto \mu_t$:** It is illustrative to contrast the path $t \mapsto \mu_t$ with the path $t \mapsto p_t$ from a recent application of Annealed LMC for sampling from the *prior* (Cordero-Encinar et al., 2025). The path $t \mapsto p_t$ can be followed efficiently because the curve $p_t$ is "continuous" in that the forward process is just an OU process with $W_2(p_t, p_{t+\delta}) \sim \delta$, resulting in an action that can be bounded. However, even when $p_t$ is close to $p_{t+\delta}$ we need not have $\mu_t$ close to $\mu_{t+\delta}$. A simple example is that of Figure 3. We have a prior represented in orange, a noisy measurement represented by the red arrow, a likelihood represented by the gray region, and a posterior represented by the blue shaded region. On the right side, the smaller mode is quite likely under the posterior. On the left side for a lower noise level, that mode has all but vanished from the posterior. This results in two distributions $\mu_t$ and $\mu_{t+\delta}$ such that $\delta$ is small, $p_t$ is close to $p_{t+\delta}$ in Wasserstein, but $\mu_t$ is not close to $\mu_{t+\delta}$. This "discontinuity" is the reason we cannot get a KL bound for $\mu_0$. However, the noising process introduces enough regularity that we can get bounds for the Wasserstein derivatives up until small $t$. Furthermore, the changes in the scores $\nabla \log \mu_{t+\Delta} - \nabla \log \mu_t$ are better behaved than changes in the log-probabilities $\log \mu_{t+\Delta} - \log \mu_t$. This allows us to get guarantees in FI rather than KL for $\mu_0$.

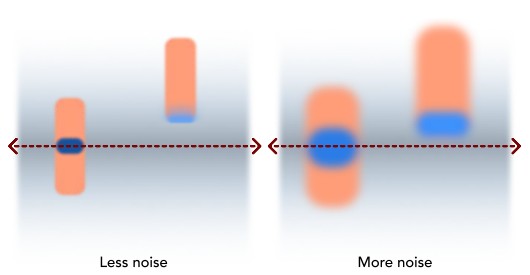

Less noise    More noise

Figure 3: "Discontinuity" of $\{\mu_t\}$: The prior consists of two vertical orange bars. We obtain a measurement, represented by the dotted line, of the vertical coordinate corrupted by some Gaussian noise. The log-likelihood is represented by the colored gradient, with dark representing regions of higher likelihood. Like the prior, the posterior represented in blue is bimodal, with one mode corresponding to each of the modes of the prior.

---

[4]This is inspired by a similar rate parameter in (Wu et al., 2024).

## 4 RESULTS

In this section, we will describe our main results. Most proofs have been deferred to the appendices, where the theorem statements contain the exact polynomial dependencies.

**Assumption 4.1.** *We make the following assumptions:*

  *(i) The prior $p_0$ is $\mathfrak{m}$-sub-Gaussian, with zero mean.*

  *(ii) the score $\nabla_x \log p_0(x)$ is $\mathfrak{L}$-Lipschitz.*

  *(iii) The log-likelihood function $R(x)$ is smooth, convex, and bounded below by $0$ such that there exists $\mathfrak{x}, \|\mathfrak{x}\| \leq \mathfrak{D}, R(\mathfrak{x}) = 0$, and $\nabla^2 R \preceq \mathfrak{R}I$.*

**Remark 4.2.** *The first assumption is generally satisfied by natural distributions, for instance, by images where each pixel is bounded intensity. The second assumption is standard in the literature (Chen et al., 2023; Lee et al., 2023). The third assumption establishes a regularity for the likelihood. In the case of noisy linear measurements $y = Ax + \sigma\eta$ for $\eta \sim \gamma$, $\mathfrak{R} \leq \|A\|^2/\sigma^2$.*

**Warm Start:** We begin by getting a sample from (close to) the limiting distribution $\mu_\infty = \lim_{t\to\infty} \mu_t$. We incur errors because we stop in finite time, and due to discretizations.

**Lemma 4.3.** *Take $T = \mathcal{O}(\frac{d}{\epsilon^2} \log \frac{\mathsf{KL}(\gamma\|\mu_\infty)}{\epsilon})$ and $T_{ws} = \mathcal{O}\left(\log \frac{d}{\epsilon}\right)$. The **Warm Start** phase of Algorithm 1 results in a sample $X_T$ satisfying $\mathsf{KL}(\mu_{T_{ws}}\|Law(X_T)) \leq \epsilon$.*

*Proof Sketch.* The Warm Start phase is LMC for the target $\mu_\infty$. Because $\gamma$ is strongly log-concave, $R$ is convex, $\gamma e^{-R}$ is strongly log-concave, so efficient sampling is possible. We can shift the guarantee to $\mu_{T_{ws}}$ because $\mu_\infty \approx \mu_{T_{ws}}$. $\qquad\square$

**Annealing Phase:** We can now begin our annealing towards the target distribution. If we traverse the annealed path $\mu_t \propto p_t e^{-R}$, the $\mathsf{KL}$ divergence between the law of the iterates $\rho_{t\kappa/\delta}$ and $\mu_t$ is

$$\mathsf{KL}\left(\mu_t\|\rho_{t\kappa/\delta}\right) \lesssim \mathsf{KL}\left(\mu_{T_{ws}}\|\rho_{T_{ws}\kappa/\delta}\right) + \mathcal{O}\left(\int_t^{T_{ws}} \|v_t\|^2 \, dt/\kappa\right),$$

where $v_t$ denotes the velocity field that generates the path $\{\mu_t\}$. An important aspect of this phase is the rate $1/\kappa$ which slows traversal of the path $\{\mu_t\}$ allowing the iterates to better track the distribution.

**Theorem 4.4.** *Suppose we run **Warm Start** phase with $T = \mathcal{O}\left(d\kappa \log(\kappa\mathsf{KL}(\gamma\|\mu_\infty))\right), T_{ws} = \log \kappa d$, following which we run the **Annealing Phase** with $\delta = \kappa^{-1/4}$. This results in a $\tau = \kappa^{-3/16}$ satisfying*

$$\mathsf{KL}\left(\mu_\tau\|\rho_{\tau\kappa/\delta}\right) \leq poly(d, 1/\kappa) \tag{3}$$

*Proof Sketch.* Important technical tools we use are bounds on the magnitude of the derivatives $\partial_t \log p_t, \partial_t \log \mu_t$ (Lemmas C.5 and C.6). These, together with Lemma B.2, allow us to bound the metric derivative $\|v_t\|^2_{L_2(\mu_t)} = \lim_{\Delta\to 0} W_2(\mu_{t+\Delta}, \mu_t)/\Delta$, where $v_t$ is the drift implementing the path $\mu_t$. The dominant term in the $\mathsf{KL}$ distance comes from the action $\int \|v_t\|^2_{L_2(\mu_t)} \, dt$. $\qquad\square$

Theorem 4.4 shows that we can track the annealed path up until $\tau$ defined above for a polynomial run time. Beyond that, $\rho_t$ does not track $\mu_{t\delta/\kappa}$ closely. We now consider the Fisher Divergence.

**Theorem 4.5.** *Suppose we run **Warm Start** phase with $T = \mathcal{O}\left(d^3\kappa \log(\kappa\mathsf{KL}(\gamma\|\mu_\infty))\right), T_{ws} = \log \kappa d$, following which we run the **Annealing Phase** with $\delta = \kappa^{-1/4}$. This results in a $\tau = \kappa^{-3/16}$ satisfying*

$$\mathsf{FI}\left(\rho_{\tau\kappa/\delta}, \mu_0\right) \leq \mathcal{O}\left(d^{3/2}\kappa^{-3/32}\right).$$

*Proof Sketch.* Consider $\partial_t \rho_t = \nabla \cdot (\rho_t \nabla \log \frac{\rho_t}{\mu_{i\delta/\kappa}})$. de Bruijn's identity states:

$$-\partial_t\mathsf{KL}\left(\rho_t\|\mu_{i\delta/\kappa}\right) \geq \mathsf{FI}\left(\rho_t, \mu_{i\delta/\kappa}\right)$$

Since we are using an annealed LMC, to telescope this as in the LMC analysis we also need to bound

$$\mathsf{KL}\left(\rho_{i\delta}\|\mu_{i\delta/\kappa}\right) - \mathsf{KL}\left(\rho_{i\delta}\|\mu_{(i-1)\delta/\kappa}\right) = -\mathbb{E}_{\rho_{i\delta}}(\log \mu_{i\delta/\kappa} - \log \mu_{(i-1)\delta/\kappa}).$$

Because the initialization $\rho_{T_{ws}}$ is sub-Gaussian, we can bound the drifts of our algorithm to show that the resulting $\rho_t$ is sub-Gaussian. Lemmas C.5 and C.6 again allow us to bound $\log \mu_{i\delta/\kappa} - \log \mu_{(i-1)\delta/\kappa}$, which we show grows at most polynomially. As a consequence, we have

$$\sum_{i=\tau\kappa/\delta}^{T_{ws}\kappa/\delta} \int_{i\delta}^{(i+1)\delta} \mathsf{FI}\big(\rho_t, \, \mu_{i\delta/\kappa}\big) \, dt \lesssim \mathsf{KL}(\rho_{T_{ws}} \| \mu_{T_{ws}})$$

From here, we finish using a weak triangle inequality for $\mathsf{FI}$ to get a guarantee against $\mu_0$.

$\square$

These results are driven by Lemmas C.5 and C.6, which effectively show that the posteriors $\mu_t$ change in a relatively mild way until some small $t > 0$, allowing us to anneal our samples in polynomial time. Putting these together, we have the following conclusion, which states that *there is an iterate close to the last iterate that satisfies a simultaneous "global" $\mathsf{KL}$ guarantee to a posterior for a noised prior and a "local" $\mathsf{FI}$ guarantee to the true posterior.*

[KL + FI] In algorithm 1, suppose we run **Warm Start** phase with $T = \mathcal{O}\left(d^3\kappa \log(\kappa \mathsf{KL}(\gamma\|\mu_\infty))\right)$, $T_{ws} = \log \kappa d$, following which we run the **Annealing Phase** with $\delta = \kappa^{1/4}$, then there is $\tau \leq \tilde{\mathcal{O}}(\kappa^{-3/16})$, such that $\rho_{\tau\kappa/\delta}$ simultaneously satisfies

- $\mathsf{KL}\big(\mu_\tau \| \rho_{\tau\kappa^{5/4}}\big) \leq \mathcal{O}(d\kappa^{-1/2})$, which implies $\mathsf{TV}\big(\rho_{\tau\kappa^{5/4}}, \, \mu_\tau\big) \leq \mathcal{O}(\sqrt{d\kappa^{-1/2}}$.
- $\mathsf{FI}\big(\rho_{\tau\kappa^{5/4}}, \, \mu_0\big) \leq \mathcal{O}(d\kappa^{-1/16})$

For this choice of $\kappa$, the algorithm has run time $\tilde{\mathcal{O}}(\kappa^{5/4})$.

## 4.1 LOCAL AND GLOBAL GUARANTEES - THE IMPLICATIONS OF COROLLARY 4

It is possible to *just* get convergence in $\mathsf{FI}$, indeed running LMC towards the posterior,

$$X_{i+1} = X_i + \delta \nabla \log \mu_0(X_i) + \sqrt{2\delta}\epsilon, \qquad \epsilon \sim \mathcal{N}(0, I),$$

results in polynomial convergence to $\mu_0$ in $\mathsf{FI}$ as in Balasubramanian et al. (2022). However, convergence in $\mathsf{FI}$ is susceptible to the phenomenon of "mode collapse", where for instance, in a multimodal distribution, the sampler significantly under-samples a specific mode depending on initialization. This is particularly critical in our setting - one could interpret posterior sampling for multi-modal priors as equivalent to conditionally sampling from a subset of modes that is consistent with a measurement. We will illustrate this below for a mixture of two Gaussians, and show how Theorem 4.4 avoids this failure mode.

Let us define a bimodal prior and a likelihood:

$$p_0 = \frac{1}{2}\mathcal{N}(\mathbf{0}, I) + \frac{1}{2}\mathcal{N}\left(\lambda \begin{bmatrix} 1 \\ 1 \end{bmatrix}, I_2\right), \qquad R(\mathbf{x}) = \frac{1}{2\eta}\|\mathrm{diag}([0, 1])\mathbf{x}\|^2$$

Let $\frac{1}{\eta'} = 1 + \frac{1}{\eta}$, and let $A_\square = \mathrm{diag}([1, \square])$ for any $\square$. Then the posterior can be written as

$$\mu_0 = \alpha_0 \mathcal{N}(\mathbf{0}, A_{\eta'}) + (1 - \alpha_0)\mathcal{N}\left(\lambda A_{\eta'}\begin{bmatrix} 1 \\ 1 \end{bmatrix}, A_{\eta'}\right), \qquad \alpha_0 = \frac{1}{1 + e^{-\frac{\lambda^2}{1+\eta}}}$$

However, we see in Lemma D.2 that even the distribution (with equal mode weights)

$$\mu_0' = \frac{1}{2}\mathcal{N}(\mathbf{0}, A_{\eta'}) + \frac{1}{2}\mathcal{N}\left(\lambda A_{\eta'}\begin{bmatrix} 1 \\ 1 \end{bmatrix}, A_{\eta'}\right)$$

satisfies $\mathsf{FI}(\mu_0, \mu_0') \leq e^{-\lambda^2\left(\frac{\eta-15}{8(1+\eta)}\right)}$. So for $\eta > 15$, $\lambda \to \infty$, $\mathsf{FI}$ completely fails to discriminate the distribution with the correct mode weights of $(\alpha_0, 1 - \alpha_0)$ from an incorrect distribution with equal weights $(1/2, 1/2)$. Now consider a *noisy* prior, and the corresponding posterior

$$p_t = \frac{1}{2}\mathcal{N}(\mathbf{0}, I) + \frac{1}{2}\mathcal{N}\left(\lambda e^{-t}\begin{bmatrix} 1 \\ 1 \end{bmatrix}, I\right), \qquad \mu_t = \alpha_t \mathcal{N}(\mathbf{0}, A_\eta) + (1 - \alpha_t)\mathcal{N}\left(e^{-t}A_\eta \mathbf{e}, A_\eta\right)$$

$$\text{with } \alpha_t = \frac{1}{1 + e^{-\frac{\lambda^2 e^{-2t}}{1+\eta}}}$$

As we saw previously, with the FI guarantee alone, there is no guarantee on the weight $\alpha$, which could range from $^1/_2$ to exponentially close to 1. However the KL (which implies a TV) guarantee shows that the weights can themselves not be off by more $\sqrt{\epsilon}$, which means $\alpha = \alpha_t \pm \sqrt{\epsilon}$.

We can now complete the discussion of Section 2.2. We saw in Section 2.1 that a FI gurarantee can be be interpreted as a type of "local" KL guarantee, and that these local guarantees cannot be stitched to get a KL guarantee. In a multimodal setting, such as this one, however, the weights of the modes themselves fall under the purview of the overall KL bound (Theorem 4.4), which sets them by solving a "simplified" posterior sampling problem.

**Remark 4.6.** *Approximating the posterior of a noised prior is in some sense the best we can do tractably. Consider the lower bound instance of Gupta et al. (2024). In summary, they use a one way function $f \colon \{-1, 1\}^d \to \{-1, 1\}^d$ such that $f(x) = y$ is easy to compute, but $f^{-1}(y) = x$ is difficult. They construct a posterior sampling problem, where the prior corresponds to a uniform distribution over $\{-1, 1\}^d$, the measurement is a specific $f(x) = y$, and the posterior would correspond to distribution concentrated on the true inverse $f^{-1}(y)$. Using the same measurement but noising the prior sufficiently results in a distribution for $x$ that is uniform over $\{-1, 1\}^d$. In other words, the posterior $\mu_t$ is concentrated on the true $f^{-1}(y)$ only for very small values of $t$.*

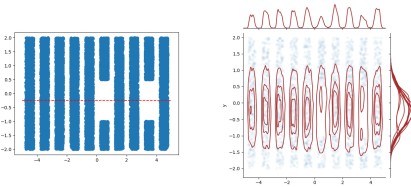

Figure 4: Our prior (shown on the left) consists of several vertical bars, two of which have gaps in them. The measurement model encourages the vertical coordinate to be $-0.25$, as indicated by the red horizontal line. The distribution of the sampler is depicted with kernel density plots for each of the resulting modes (shown to the right in red overlaid on top of the prior).

**Remark 4.7.** *Consider a prior of several vertical "bars" in $\mathbb{R}^2$, two of which have a gap in them in some range of the vertical coordinate (see Figure 4). Our measurement operator gives us only a noisy measurement of the vertical coordinate (the red dotted line, in this case at $y = -.25$). The two bars with gaps in them should be very unlikely under the true posterior. However, the posterior of a* noised *prior would not notice this gap for some time. The annealed Langevin algorithm we describe results in the sampler shown on the right. A kernel density estimate for each of the resulting modes is plotted in red. Note that each of the modes is discovered, and as we see from the marginals, the two modes that should have a lower weight under the posterior do have a smaller weight.*

## 5 CONCLUSION

We study the Annealed Langevin Monte Carlo algorithm to generate samples from an approximation to the true posterior distribution. We show that this algorithm simultaneously satisfies two properties: when initialized with an efficient "warm-start", an iterate close to the final iterate is *(i)* close in KL with respect to the posterior with a noisy prior, and *(ii)* close in FI with respect to the true posterior. To the best of our knowledge, these constitute the first polynomial-time results for a suitable notion of approximate posterior sampling.

We believe this type of guarantee is also possible with other popular posterior sampling frameworks like Split-Gibbs sampling, which can be interpreted as a different discrete path through the space of distributions. Furthermore, there may be other paths $\{\mu_t\}$ that allow us to sample from interpretable approximations to the true posterior (such as on that more closely aligns with DDPM, rather than Annealed Langevin); this is an interesting avenue for future work.

## ACKNOWLEDGMENTS

This research has been supported by NSF Grants 2019844, 2505865 and 2112471, and the UT Austin Machine Learning Lab.

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

# A PRELIMINARIES

## A.1 NOTATION AND OVERVIEW

**Notation.** The prior is denoted $p$. The log-likelihood, or the measurement consistency, is denoted $R$. We denote by $p_t$ the distribution $p$ passed through the OU channel, which is to say, if $X_t$ is an OU process with $X_0$ having law $p$, then $p_t$ is the law of $X_t$. We use $\mu$ to denote posteriors, so $\mu_0$ is the posterior $p_0 e^{-R}/Z$, and $\mu_t$ is $p_t e^{-R}$. We use $\circ$ to denote composition, so $(f \circ g)(x) = f(g(x))$

We use $C_c^\infty(\mathcal{U})$ to denote the space of all smooth functions on $\mathcal{U}$ with compact support, $\mathcal{P}_2(\mathbb{R}^d)$ to denote the set of measures on $\mathbb{R}^d$, and $\mathcal{P}_{2,ac}(\mathbb{R}^d)$ to denote the set of measures that are absolutely continuous with respect to the Lebesgue measure.

**Remark A.1** (Constants greater than one). *For simplicity, we assume that each of the constants defined in Assumption 4.1 is a constant greater than one.*

**Overview.** In Section A.2 we review some identities that will be useful. In A.3 we state some prior work with references. In Appendix B we discuss various aspects of the algorithm discussed in Section 3. In Appendix C we state and prove some bounds that are useful to Appendix B.

## A.2 PRELIMINARIES

**Lemma A.2** (Identities). *We have the following identities, under benign regularity conditions. These are commonly used in the literature but are repeated here for completeness*

1. *For $f, g : \mathbb{R}^d \to \mathbb{R}$, we have $\nabla \cdot (f * g) = (\nabla \cdot f) * g$*

2. *For $f : \mathbb{R}^d \to \mathbb{R}^d, g : \mathbb{R}^d \to \mathbb{R}$, we have $\nabla(f * g) = (\nabla f) * g$*

3. *For $f, g : \mathbb{R}^d \to \mathbb{R}$, we have $\Delta(f * g) = (\Delta f) * g$*

4. *For $f : \mathbb{R} \to \mathbb{R}, g : \mathbb{R}^d \to \mathbb{R}, \nabla \cdot (f \nabla g) = \nabla f \cdot \nabla g + f \Delta g$*

5. *For $f : \mathbb{R} \to \mathbb{R}, f \nabla \log f = \nabla f$*

*Proof.* Follows from switching the order of the integrals and the derivatives. The principle is that convolution commutes with linear operators.

1.
$$\nabla \cdot (f * g) = \sum_i \partial_i \int f(x - y)g(y) \, dy = \int \sum_i \partial_i \left( f(x-y)g(y) \right) dy$$
$$= \int \sum_i \left( \partial_i f(x - y) \right) g(y) dy = (\nabla \cdot f) * g$$

2.
$$\nabla(f * g) = \nabla_x \int f(x - y)g(y) \, dy = \int \nabla_x f(x - y)g(y) dy = (\nabla f) * g$$

3. Follows from the above two:
$$\Delta(f * g) = \nabla \cdot \nabla(f * g) = \nabla \cdot ((\nabla f) * g) = \nabla \cdot (\nabla f) * g = (\Delta f) * g$$

The remaining are common calculus manipulations. $\square$

**Lemma A.3** (Gaussians). *The following hold for Gaussians $\gamma_{\sigma^2}(x)$*

1. $\nabla \gamma_{\sigma^2} = -\frac{x}{\sigma^2} \gamma_{\sigma^2}$

2. $\Delta \gamma_{\sigma^2} = \left( \frac{\|x\|}{\sigma^4} - \frac{d}{\sigma^2} \right) \gamma_{\sigma^2}$

3. $\Delta \log \gamma = -\frac{d}{\sigma^2}$

The above also follow from standard calculus rules.

### A.3 MISCELLENEOUS RESULTS

**Lemma A.4** (Girsanov, (Øksendal, 2003)). *Let $X_0 \sim \rho_0, X_0' \sim \rho_0'$, and suppose*

$$dX_t = v_t(X_t)\, dt + \sqrt{2}\, dB_t \iff \partial_t \rho_t = -\nabla \cdot (\rho_t v_t) + \Delta \rho_t$$
$$dX_t' = v_t'(X_t')\, dt + \sqrt{2}\, dB_t \iff \partial_t \rho_t' = -\nabla \cdot (\rho_t' v_t') + \Delta \rho_t' \tag{4}$$

*The KL divergence between $\rho_t$ and $\rho_t'$ can be bounded as*

$$KL(\rho_t \| \rho_t') \leq KL(\rho_0 \| \rho_0') + \frac{1}{4}\mathbb{E}_{\{X_t\}} \int_0^T \|v_t(X_t) - v_t'(X_t)\|^2\, dt$$

**Lemma A.5** (LMC convergence under Log-Concavity (Vempala & Wibisono, 2022)). *Let $k \in \mathbb{N}$, and let $\mu_{kh}$ denote the law of the $k$-th iterate of the Langevin Monte Carlo (LMC) algorithm with step size $h > 0$. Assume that the target distribution $\pi \propto \exp(-V)$ satisfies a logarithmic Sobolev inequality with constant $C_{LSI}(\pi) \leq \frac{1}{\alpha}$, and that $\nabla V$ is $\beta$-Lipschitz. Then, for all $h \leq \frac{1}{4\beta}$ and for all $N \in \mathbb{N}$,*

$$\mathrm{KL}(\mu_{Nh} \| \pi) \leq \exp(-\alpha Nh)\, \mathrm{KL}(\mu_0 \| \pi) + \mathcal{O}\left(\frac{\beta^2 dh}{\alpha}\right).$$

*In particular, letting $\kappa := \frac{\beta}{\alpha}$, for all $\varepsilon \in [0, \kappa\sqrt{d}]$ and for step size $h \asymp \frac{\varepsilon^2}{\beta\kappa d}$, we have $\sqrt{\mathrm{KL}(\mu_{Nh} \| \pi)} \leq \epsilon$ after $N = \mathcal{O}\left(\frac{\kappa^2 d}{\epsilon^2} \log \frac{\mathrm{KL}(\mu_0 \| \pi)}{\epsilon^2}\right)$ iterations.*

**Lemma A.6** (HWI inequality (Otto & Villani, 2000)). *Let $\pi \in \mathcal{P}_2(\mathbb{R}^d)$ be a reference measure, and let $\rho \in \mathcal{P}_2(\mathbb{R}^d)$. We have*

$$KL(\pi \| \rho) \leq W_2(\pi, \rho)\sqrt{FI(\pi, \rho)}$$

**Lemma A.7** (Talagrands transportation inequality (Chewi, 2023)). *Let $\pi \in \mathcal{P}_2(\mathbb{R}^d)$ be $\alpha-$strongly concave. Then we have*

$$KL(\rho \| \pi) \geq \frac{\alpha}{2}W_2^2(\rho, \pi).$$

## B PROOFS FOR ANNEALED LANGEVIN

In this section, we elaborate on the proofs of section 3. Recall our general strategy for sampling. We

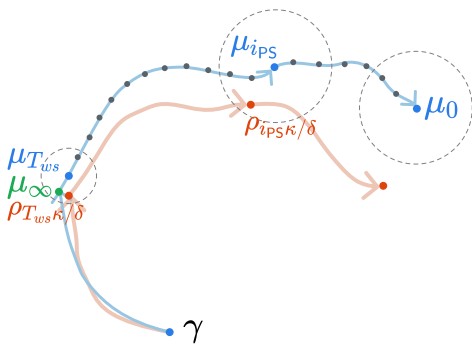

Figure 5: (1.) We sample using LMC from $\mu_T \approx \mu_\infty$. (2.) We run Annealed LMC along the path $t \mapsto \mu_t$.

begin by showing that the limiting distribution exists $\lim_{t \to \infty} \mu_t = \mu_\infty$.

**Lemma B.1.** *Let $\mu_t = p_t e^{-R}/Z$. The sequence $\mu_t$ converges weakly to $\mu_\infty = \gamma e^{-R}/Z$.*

*Proof.* First note that if $p \in C_c^\infty(\mathbb{R})$, then $\lim_{t\to\infty} e^{td} p(e^t x) = \delta$ in the sense of distributions. We need to show for every $\phi \in C_c^\infty(\mathbb{R})$ that $\mathbb{E}_{\mu_\infty}\phi = \lim_{t\to\infty} \mathbb{E}_{\mu_t}\phi$. We have

$$
\begin{aligned}
\lim_{t\to\infty} \mathbb{E}_{\mu_t}\phi &= \lim_{t\to\infty} \frac{\int \phi(x)e^{-R(x)}p_t(x)\, dx}{\int e^{-R(x)}p_t(x)\, dx} \\
&= \frac{\lim_{t\to\infty} \int \phi(x)e^{-R(x)}p_t(x)\, dx}{\lim_{t\to\infty} \int e^{-R(x)}p_t(x)\, dx} \\
&= \frac{\int \lim_{t\to\infty} \phi(x)e^{-R(x)}p_t(x)\, dx}{\int \lim_{t\to\infty} e^{-R(x)}p_t(x)\, dx} \\
&= \frac{\int \phi(x)e^{-R(x)}\gamma(x)\, dx}{\int e^{-R(x)}\gamma(x)\, dx} \\
&= \mathbb{E}_{\mu_\infty}\phi
\end{aligned}
$$

The second equality holds as long as $\lim_{t\to\infty} \int e^{-R(x)} \left( \int e^{td}p(e^t(x-y))\gamma_{1-e^{-2t}}(y)\, dy \right)\, dx \neq 0$. The third requires dominated convergence for $p_t(x)e^{-R(x)}\phi(x)$ and $p_t(x)e^{-R(x)}$. The fourth requires $\lim_{t\to\infty} p_t = \gamma$. We will confirm these below in reverse order. First we have

$$
\begin{aligned}
\lim_{t\to\infty} p_t &= \lim_{t\to\infty} \int e^{td}p(e^t(x-y))\gamma_{1-e^{-2t}}(y)\, dy \\
&= \int \lim_{t\to\infty} \left( e^{td}p(e^t(x-y))\gamma_{1-e^{-2t}}(y) \right)\, dy \\
&= \int \left( \lim_{t\to\infty} e^{td}p(e^t(x-y)) \right)\left( \lim_{t\to\infty} \gamma_{1-e^{-2t}}(y) \right)\, dy \\
&= \int \delta(x-y)\gamma(y)\, dy = \gamma
\end{aligned}
$$

From C.4, we know $p_t e^{-R(x)}\phi(x) \leq \frac{1}{(1-e^{-2t})^{d/2}} e^{-R(x)}\phi(x)$ pointwise, and $\int \frac{1}{(1-e^{-2t})^{d/2}} e^{-R(x)}\phi(x)\, dx = \frac{1}{(1-e^{-2t})^{d/2}} \int e^{-R(x)}\phi(x)\, dx$. Because $e^{-R}$ and $\phi$ are both square integrable, $e^{-R}\phi$ is integrable from Cauchy Schwartz, and we can use the dominated convergence theorem to show that

$$
\lim_{t\to\infty} \int e^{-R(x)}p_t(x)\phi(x)\, dx = \int \lim_{t\to\infty} e^{-R(x)}p_t(x)\phi(x)\, dx.
$$

We can show similarly that

$$
\lim_{t\to\infty} \int e^{-R(x)}p_t(x)\, dx = \int \lim_{t\to\infty} e^{-R(x)}p_t(x)\, dx.
$$

Finally, we have

$$
\lim_{t\to\infty} \int e^{-R(x)}p_t(x)\, dx = \int \lim_{t\to\infty} e^{-R(x)}p_t(x)\, dx = \int e^{-R(x)}\gamma(x)\, dx > 0.
$$

$\square$

This distribution is log-concave, and we can show that LMC converges quickly to $\mu_\infty$. Let $\mathsf{Law}(X_t)$ denote the law of $X_t$ when $X_0 \sim \gamma$ and we run LMC towards $\mu_\infty$ for time $T$ (Line 4 of Algorithm 1). We show that $\rho_{ws} \approx \mu_\infty \approx \mu_{T_{ws}}$ for sufficiently large $T_{ws}, T$. The standard results on LMC convergence are usually given in terms of the $\mathsf{KL}$ divergence between the law of the iterate and the target distribution. To apply Girsanov's Theorem A.4 later in 4.4 we need the $\mathsf{KL}$ divergence between the target and the law of the iterate.

**Lemma 4.3.** *Take $T = \mathcal{O}(\frac{d}{\epsilon^2} \log \frac{\mathsf{KL}(\gamma \| \mu_\infty)}{\epsilon})$ and $T_{ws} = \mathcal{O}\left( \log \frac{d}{\epsilon} \right)$. The **Warm Start** phase of Algorithm 1 results in a sample $X_T$ satisfying $\mathsf{KL}(\mu_{T_{ws}} \| \mathsf{Law}(X_T)) \leq \epsilon$.*

*Proof.* We will do this in three steps. First, we will show that standard results in this setting bound $\mathsf{KL}(\mathsf{Law}(X_T) \| \mu_\infty)$. Then we will bound $\mathsf{KL}(\mu_\infty \| \mathsf{Law}(X_T))$ from $\mathsf{KL}(\mathsf{Law}(X_T) \| \mu_\infty)$. In

general, we cannot reverse the order of the arguments in a $\mathsf{KL}$ divergence but we can under some conditions (log-concavity + lipschitzness of the scores + subgaussian target), and then show that $\mathsf{KL}(\mu_{T_{ws}}\|\mathsf{Law}(X_T))$ is small.

**Step 1. Showing that $\mathsf{KL}(\mathsf{Law}(X_T)\|\mu_\infty) < \epsilon$**

The drift term

$$\nabla \log \mu_\infty = \nabla \log(\gamma e^{-R}/Z) = -x - \nabla R$$

satisfies

$$\|\nabla(-x - \nabla R)\| \leq \sqrt{d} + \|\nabla^2 R\| \leq \sqrt{d} + \mathfrak{R},$$

and also $\|\nabla(x - \nabla R)\| \geq d$ from convexity of $R$, so $\mu_\infty$ is $d-$log-concave. From Lemma A.5 (which is from Vempala & Wibisono (2022)), we see that we can take $\beta = 1 + \mathfrak{R}$, $\alpha = 1 + \mathfrak{R}$, $\delta \asymp \frac{\epsilon^2}{(1+\mathfrak{R})d}$ and to get that at $T = \mathcal{O}\left(\frac{d}{\epsilon^2} \log \frac{\mathsf{KL}(\gamma\|\mu_\infty)}{\epsilon^2}\right)$ iterations we have $\mathsf{KL}(\mathsf{Law}(X_T)\|\mu_\infty) \leq \epsilon^2$.

**Step 2. Showing that $\mathsf{KL}(\mu_\infty\|\mathsf{Law}(X_T)) < \epsilon$.**

By Lemma A.6 we have

$$\mathsf{KL}(\mu_\infty\|\mathsf{Law}(X_T)) \leq W_2(\mathsf{Law}(X_T), \mu_\infty)\sqrt{\mathsf{FI}(\mu_\infty, \mathsf{Law}(X_T))}.$$

The Fisher divergence is bounded by a dimension dependent constant

$$\begin{aligned}
\mathsf{FI}(\mu_\infty, \mathsf{Law}(X_T)) &= \mathbb{E}_{\mu_\infty} \|\nabla \log \mu_\infty - \nabla \log \mathsf{Law}(X_T)\|^2 \\
&\leq 2\mathbb{E}_{\mu_\infty} \|\nabla \log \mu_\infty\|^2 + 2\mathbb{E}_{\mu_\infty} \|\nabla \log \mathsf{Law}(X_T)\|^2 \\
&\leq \text{poly}(\mathfrak{m}, \mathfrak{R}, \mathfrak{L}, d)
\end{aligned}$$

Overall we get $\mathsf{KL}(\mu_\infty\|\mathsf{Law}(X_T)) \leq \text{poly}(\mathfrak{m}, \mathfrak{R}, \mathfrak{L})W_2(\mathsf{Law}(X_T), \mu_\infty)$.

Note that $\mu_\infty$ is at least $1-$strongly log-concave, so we have from Talagrands transportation inequality A.7

$$\begin{aligned}
\mathsf{KL}(\mu_\infty\|\mathsf{Law}(X_T)) &\leq \text{poly}(\mathfrak{m}, \mathfrak{R}, \mathfrak{L}) \, W_2(\mathsf{Law}(X_T), \mu_\infty) \\
&\leq \text{poly}(\mathfrak{m}, \mathfrak{R}, \mathfrak{L})\sqrt{\mathsf{KL}(\mathsf{Law}(X_T)\|\mu_\infty)} \leq \text{poly}(\mathfrak{m}, \mathfrak{R}, \mathfrak{L}) \, \epsilon
\end{aligned}$$

**Step 3. Showing that $\mathsf{KL}(\mu_{T_{ws}}\|\mathsf{Law}(X_T)) < \epsilon$**

We can now also show that $\mathsf{KL}(\rho_{T_{ws}}\|\mu_{T_{ws}})$ is small

$$\begin{aligned}
\mathsf{KL}(\mu_{T_{ws}}\|\mathsf{Law}(X_T)) &= \mathbb{E}_{\mu_{T_{ws}}} \log \mu_{T_{ws}} - \log \mathsf{Law}(X_T) \\
&= \mathbb{E}_{\mu_{T_{ws}}} \log \mu_{T_{ws}} - \log \mu_\infty + \log \mu_\infty - \log \mathsf{Law}(X_T) \\
&= \mathsf{KL}(\mu_{T_{ws}}\|\mu_\infty) + \mathbb{E}_{\mu_{T_{ws}}} (\log \mu_\infty - \log \mathsf{Law}(X_T)) \\
&= \mathbb{E}_{\mu_\infty} (\log \mu_\infty - \log \mathsf{Law}(X_T)) \frac{\mu_{T_{ws}}}{\mu_\infty} \\
&\leq \mathbb{E}_{\mu_\infty} [(\log \mu_\infty - \log \mathsf{Law}(X_T))] \sup_x \frac{\mu_{T_{ws}}(x)}{\mu_\infty(x)} \\
&= \mathsf{KL}(\mu_\infty\|\mathsf{Law}(X_T)) \sup_x \frac{\mu_{T_{ws}}(x)}{\mu_\infty(x)} \\
&= \mathsf{KL}(\mu_\infty\|\mathsf{Law}(X_T)) \, e^{\sup_x |\log \mu_{T_{ws}} - \log \mu_\infty|}
\end{aligned}$$

We have from Lemma C.7

$$e^{\sup_x |\log \mu_{T_{ws}} - \log \mu_\infty|} \leq e^{\frac{e^{-2T_{ws}}}{1 - e^{-2T_{ws}}} \text{poly}(\mathfrak{m}, \mathfrak{L}, \mathfrak{R}, d)}$$

So if we set $T_{ws} = \mathcal{O}(\log \frac{d}{\epsilon})$, we get $\mathsf{KL}(\mathsf{Law}(X_T)\|\mu_{T_{ws}}) < \text{poly}(\mathfrak{m}, \mathfrak{R}, \mathfrak{L})\epsilon$. $\qquad \square$

A map $t \mapsto \pi_t$ from $[0, T] \rightarrow \mathcal{P}_2(\mathbb{R}^d)$ is *absolutely continuous* if for all $t$,

$$|\dot{\mu}(t)| := \lim_{\delta \to 0} \frac{W_2(\mu_t, \mu_{t+\delta})}{\delta} < \infty.$$

Consider the continuity equation $\partial \pi_t = -\nabla \cdot (\pi_t v_t)$. Any choice of $v_t$ results in a curve $t \mapsto \pi_t$, but, conversely if $t \mapsto \pi_t$ is an absolutely continuous curve, there exists a choice of $v_t$, such that $\partial_t \pi_t = -\nabla \cdot (\pi_t v_t)$ and $\|v_t\|_{L_2(\pi_t)} \leq |\dot{\mu}(t)|$. We refer the reader to Chewi (2023) or Ambrosio et al. (2008) for a more elaborate exposition. In order to use Girsanov's Theorem to bound the KL distance for the drift between the target and the law of the iterate during annealed LMC, we will need to bound this derivative $|\dot{\mu}(t)|$.

**Lemma B.2.** *The path $t \mapsto \mu_t$ is an absolutely continuous curve. There exists a velocity field $v_t$ satisfying $\partial_t \mu_t = -\nabla \cdot (\mu_t v_t)$, and*

$$\|v_t\|_{L_2(\mu_t)} \leq \frac{de^{-t}}{(1 - e^{-2t})^4} poly(\mathfrak{m}, \mathfrak{R}, \mathfrak{L}).$$

*Proof.* We have $W_1(\mu, \nu) = \inf_{(X,Y) \sim \pi, \pi_X = \mu, \pi_Y = \nu} \int |X - Y| \, d\pi$. From duality we get the following equivalent characterization

$$W_1(\mu, \nu) = \sup \left\{ \int f \, d(\mu - \nu) \, \middle| \, \text{Lip}(f) \leq 1 \right\} \tag{5}$$

To tie this to $W_2$, recall that for all $\mathfrak{m}-$subgaussian $\mu, \nu$, we have $W_2(\mu, \nu) \leq \sqrt{\mathfrak{m}} W_1(\mu, \nu)$. Without loss of generality we can assume $f \geq 0$, because for any constant $c$, in particular for $\inf f$, we have $\int f \, d(\mu - \nu) = \int (f - c) \, d(\mu - \nu)$.

So we have

$$
\begin{aligned}
W_1(\mu, \nu) &= \sup \left\{ \int f \, d(\mu - \nu) \, \middle| \, \text{Lip}(f) \leq 1 \right\} \\
&= \sup \left\{ \int f \, d(\mu - \nu) - \int \inf f \, d(\mu - \nu) \, \middle| \, \text{Lip}(f) \leq 1 \right\} \\
&= \sup \left\{ \int f \, d(\mu - \nu) \, \middle| \, \text{Lip}(f) \leq 1, f \geq 0 \right\}
\end{aligned}
$$

Take any specific $f$. From $\text{Lip}(f) \leq 1$, we have $f \leq \|x\|$, and from Lemma C.6 we have $|\partial_t \log \mu_t| \leq \frac{e^{-t}}{(1-e^{-2t})^4} \sum_{i=0}^2 a_i \|x\|^i$ for $a_i = d \, \text{poly}(\mathfrak{m}, \mathfrak{L}, \mathfrak{R})$. Putting these together we have

$$f |\partial_t \log \mu_t| \leq \frac{e^{-t}}{(1 - e^{-2t})^4} \sum_{i=0}^2 a_i \|x\|^i.$$

From Lemmas C.1 and C.3 we have $\mathbb{E}_{\mu_t} f |\partial_t \log \mu_t| \leq \frac{e^{-t} d}{(1-e^{-2t})^4} \text{poly}(\mathfrak{m}, \mathfrak{R}, \mathfrak{L})$.

From this, we have for all Lipschitz $f$

$$\int f \, d\mu_t - d\mu_{t+\delta} = \int_t^{t+\delta} \int f \partial_t \log \mu_t \, d\mu_t \leq \frac{e^{-t} d}{(1 - e^{-2t})^4} \text{poly}(\mathfrak{m}, \mathfrak{R}, \mathfrak{L}) \delta.$$

Since this is true for all $f$, this shows uniform convergence of $\int f \, d\mu_{t+\delta}$ to $\int f \, d\mu_t$. In particular, this means it is also true of the supremum

$$\lim_{\delta \to 0} W_2(\mu_t, \mu_{t+\delta}) \leq \sqrt{\mathfrak{m}} \lim_{\delta \to 0} \frac{\sup \int f \, d(\mu_t - \mu_{t-\delta})}{\delta} = \frac{\sup \int f(\partial_t \ln \mu_t) \mu_t \, dx}{\delta} \leq \frac{e^{-t} d}{(1 - e^{-2t})^4} \text{poly}(\mathfrak{m}, \mathfrak{R}, \mathfrak{L}).$$

$\square$

**Theorem 4.5.** *Suppose we run **Warm Start** phase with $T = \mathcal{O}\left(d^3\kappa \log(\kappa \mathsf{KL}(\gamma\|\mu_\infty))\right)$, $T_{ws} = \log \kappa d$, following which we run the **Annealing Phase** with $\delta = \kappa^{-1/4}$. This results in a $\tau = \kappa^{-3/16}$ satisfying*

$$\mathsf{FI}(\rho_{\tau\kappa/\delta}, \mu_0) \leq \mathcal{O}\left(d^{3/2}\kappa^{-3/32}\right).$$

*Proof.* We use the following from Appendix C of Balasubramanian et al. (2022). We have that $\nabla \log \mu_{i\delta/\kappa}$ is $\mathfrak{L}$ Lipshitz

$$\mathsf{KL}\left(\rho_{i\delta+\delta}\|\mu_{i\delta/\kappa}\right) - \mathsf{KL}\left(\rho_{i\delta}\|\mu_{i\delta/\kappa}\right) \geq \frac{1}{2}\int_{i\delta}^{i\delta+\delta}\mathsf{FI}\left(\rho_{i\delta+\delta}, \mu_{i\delta/\kappa}\right) - 4\mathfrak{L}^2 d\delta^2$$

and

$$\mathsf{KL}\left(\rho_{i\delta}\|\mu_{(i-1)\delta/\kappa}\right) - \mathsf{KL}\left(\rho_{i\delta}\|\mu_{i\delta/\kappa}\right)$$
$$= \mathbb{E}_{\rho_{i\delta}}\log\frac{\rho_{i\delta}}{\mu_{(i-1)\delta/\kappa}} - \mathbb{E}_{\rho_{i\delta}}\log\frac{\rho_{i\delta}}{\mu_{i\delta/\kappa}} = \mathbb{E}_{\rho_{i\delta}}\log\frac{\mu_{i\delta/\kappa}}{\mu_{(i-1)\delta/\kappa}}$$

Putting these together we have

$$\mathsf{KL}\left(\rho_{(i\delta+\delta)}\|\mu_{i\delta/\kappa}\right) - \mathsf{KL}\left(\rho_{i\delta}\|\mu_{(i-1)\delta/\kappa}\right) + \mathbb{E}_{\rho_{i\delta}}\log\frac{\mu_{i\delta/\kappa}}{\mu_{(i-1)\delta/\kappa}} \geq \frac{1}{2}\int_{i\delta}^{i\delta+\delta}\mathsf{FI}\left(\rho_t, \mu_{i\delta/\kappa}\right) \, dt - 4\mathfrak{L}^2 d\delta^2$$

We can telescope this:

$$\sum_{i=i_*}^{T_{ws}\kappa/\delta}\left(\mathsf{KL}\left(\rho_{i\delta+\delta}\|\mu_{i\delta/\kappa}\right) - \mathsf{KL}\left(\rho_{i\delta}\|\mu_{(i-1)\delta/\kappa}\right) + \mathbb{E}_{\rho_{i\delta}}\log\frac{\mu_{i\delta/\kappa}}{\mu_{(i-1)\delta/\kappa}}\right)$$

$$\geq \sum_{i=i_*}^{T_{ws}\kappa/\delta}\frac{1}{2}\left(\int_{i\delta}^{i\delta+\delta}\mathsf{FI}\left(\rho_t, \mu_{i\delta/\kappa}\right) \, dt - 4\mathfrak{L}^2 d\delta^2\right)$$

$$\implies \mathsf{KL}(\rho_T\|\mu_{T-\delta}) - \mathsf{KL}\left(\rho_\delta\|\mu_{i_*\delta/\kappa}\right) + \sum_{i=i_*}^{T_{ws}\kappa/\delta}\mathbb{E}_{\rho_{i\delta}}\log\frac{\mu_{i\delta/\kappa}}{\mu_{(i-1)\delta/\kappa}}$$

$$\geq \sum_{i=i_*}^{T_{ws}\kappa/\delta}\frac{1}{2}\int_{i\delta}^{(i+1)\delta}\mathsf{FI}\left(\rho_t, \mu_{i\delta/\kappa}\right) \, dt - 4\mathfrak{L}^2 d\delta T_{ws}\kappa$$

We need to bound $\sum \mathbb{E}_{\rho_{i\delta}}\log\frac{\mu_{i\delta/\kappa}}{\mu_{(i-1)\delta/\kappa}}$. Because $\rho_{i\delta}$ is $\mathfrak{m}-$subgaussian, we have

$$\sum \mathbb{E}_{\rho_{i\delta}}\log\frac{\mu_{i\delta/\kappa}}{\mu_{(i-1)\delta/\kappa}} \leq \sum\mathbb{E}_{\rho_{i\delta}}\log\frac{\mu_{i\delta/\kappa}}{\mu_{(i-1)\delta/\kappa}}$$

$$= \sum\mathbb{E}_{\rho_{i\delta}}\int_{(i-1)\delta/\kappa}^{i\delta}\partial_t\log\mu_t \, dt \leq \sum\int_{(i-1)\delta}^{i\delta}\mathbb{E}_{\rho_{i\delta}}|\partial_t\log\mu_t| \, dt$$

$$\leq \sum\int_{(i-1)\delta/\kappa}^{i\delta/\kappa}\frac{de^{-t}}{(1-e^{-2t})^4}\mathrm{poly}(\mathfrak{m}, \mathfrak{R}, \mathfrak{L}) \, dt$$

$$= \frac{de^{-(T_{ws}\kappa/\delta)^\alpha\delta/\kappa}}{(1-e^{-2(T_{ws}\kappa/\delta)^\alpha\delta/\kappa})^4}\mathrm{poly}(\mathfrak{m}, \mathfrak{R}, \mathfrak{L})$$

so if $(T_{ws}\kappa/\delta)^\alpha\delta/\kappa < 1$:

$$\mathsf{KL}(\rho_T\|\mu_{T-\delta}) + \frac{d\,\mathrm{poly}(\mathfrak{m}, \mathfrak{R}, \mathfrak{L})}{16T_{ws}^{4\alpha}(\delta/\kappa)^{4-4\alpha}} + 4L^2 d\delta T_{ws}\kappa$$

$$\geq \sum_{i=(T_{ws}\kappa/\delta)^\alpha}^{T_{ws}\kappa/\delta}\frac{1}{2}\int_{i\delta}^{(i+1)\delta}\mathsf{FI}\left(\rho_t, \mu_{i\delta/\kappa}\right) \, dt$$

In LD, each of the $\mathsf{FI}$ are computed with respect to the target distribution, and an average iterate guarantee can be derived using the convexity of $\mathsf{FI}$ in its first argument. In our case, the second argument is changing over the course of the integral, so we need a "triangle inequality" to change the second argument to $\mu_0$. We have

$$
\begin{aligned}
\mathsf{FI}(\rho_t, \mu_0) &= \mathbb{E}_{\rho_t} \left\| \nabla \log \rho_t - \nabla \log \mu_0 \right\|^2 \\
&\leq 2\mathbb{E}_{\rho_t} \left\| \nabla \log \rho_t - \nabla \log \mu_t \right\|^2 + 2\mathbb{E}_{\rho_t} \left\| \nabla \log \mu_t - \nabla \log \mu_0 \right\|^2 \\
&\leq 2\mathsf{FI}(\rho_t, \mu_t) + 2\mathbb{E}_{\rho_t} \left\| \nabla \log p_t - \nabla \log p_0 \right\|^2 \\
&\leq 2\mathsf{FI}(\rho_t, \mu_t) + \mathrm{poly}(\mathfrak{m}, \mathfrak{L}, d) t^2
\end{aligned}
$$

We will use the bound

$$
\begin{aligned}
\sum_{i=(T_{ws}\kappa/\delta)^\alpha}^{T_{ws}\kappa/\delta} \frac{1}{2} \int_{i\delta}^{(i+1)\delta} \mathsf{FI}(\rho_t, \mu_{i\delta/\kappa}) \; dt &\geq (T_{ws}\kappa/\delta)^\alpha \min_{i \in [(T_{ws}\kappa/\delta)^\alpha, 2(T_{ws}\kappa/\delta)^\alpha]} \frac{1}{2} \int_{i\delta}^{(i+1)\delta} \mathsf{FI}(\rho_t, \mu_{i\delta/\kappa}) \; dt \\
&\geq (T_{ws}\kappa/\delta)^\alpha \min_{i \in [(T_{ws}\kappa/\delta)^\alpha, 2(T_{ws}\kappa/\delta)^\alpha]} \min_{t \in [i\delta, i\delta+\delta]} \frac{\delta}{2} \mathsf{FI}(\rho_t, \mu_{i\delta/\kappa})
\end{aligned}
$$

to get that there exists $\tau \in [(T_{ws}\kappa/\delta)^\alpha, 2(T_{ws}\kappa/\delta)^\alpha]$ such that

$$
\mathsf{FI}(\rho_\tau, \mu_{i\delta/\kappa}) \leq \frac{\mathsf{KL}(\rho_{T_{ws}\kappa} \| \mu_{T_{ws}}) + \frac{\mathrm{poly}(\mathfrak{m}, \mathfrak{R}, \mathfrak{L})}{T_{ws}^{4\alpha}(\delta/\kappa)^{4-4\alpha}} + 4\mathfrak{L}^2 d\delta T_{ws}\kappa}{\delta(T_{ws}\kappa/\delta)^\alpha}
$$

From our approximate triangle inequality for $\mathsf{FI}$, we have that there exists $\tau \in [\delta(T_{ws}\kappa/\delta)^\alpha, 2\delta(T_{ws}\kappa/\delta)^\alpha]$ such that

$$
\mathsf{FI}(\rho_\tau, \mu_0) \leq 2\mathsf{FI}(\rho_\tau, \mu_{i\delta/\kappa}) + d\,\mathrm{poly}(\mathfrak{m}, \mathfrak{L})(T_{ws}\kappa/\delta)^{2\alpha}\delta^2/\kappa^2
$$

Suppose the warm start phase is run such that $\mathsf{KL}(\mu_{T_{ws}} \| \mathrm{Law}(X_T)) \leq \epsilon_{ws}$ (recall that this takes time $\mathrm{poly}(1/\epsilon_{ws})$).

$$
\begin{aligned}
\mathsf{FI}(\rho_\tau, \mu_0) &\leq \frac{\epsilon_{ws}}{T_{ws}^\alpha \kappa^\alpha \delta^{1-\alpha}} + \frac{d\kappa^{4-5\alpha}}{T_{ws}^{5\alpha}\delta^{5-5\alpha}}\mathrm{poly}(\mathfrak{m}, \mathfrak{R}, \mathfrak{L}) + d\delta^\alpha T_{ws}^{1-\alpha}\kappa^{1-\alpha}\,\mathrm{poly}(\mathfrak{m}, \mathfrak{R}, \mathfrak{L}) \\
&\quad + d\,\mathrm{poly}(\mathfrak{m}, \mathfrak{L})\,T_{ws}^{2\alpha}\kappa^{2\alpha-2}\delta^{2-2\alpha}
\end{aligned}
$$

If we take $\kappa = \delta^{-4}$, we have

$$
\mathsf{FI}(\rho_\tau, \mu_0) \leq \frac{\epsilon_{ws}}{T_{ws}^\alpha \kappa^{\frac{5\alpha-1}{4}}} + d\left( \frac{\kappa^{\frac{21-25\alpha}{4}}}{T_{ws}^{5\alpha}} + T_{ws}^{1-\alpha}\kappa^{\frac{4-5\alpha}{4}} + T_{ws}^{2\alpha}\kappa^{\frac{5\alpha-5}{2}} \right)\mathrm{poly}(\mathfrak{m}, \mathfrak{R}, \mathfrak{L})
$$

Finally, setting $\alpha = {}^{17}\!/_{20}$, we have

$$
\begin{aligned}
FI\rho_\tau\mu_0 &\leq \frac{\epsilon_{ws}}{T_{ws}^\alpha \kappa^{\frac{5\alpha-1}{4}}} + d\left( \frac{\kappa^{\frac{21-25\alpha}{4}}}{T_{ws}^{5\alpha}} + T_{ws}^{1-\alpha}\kappa^{\frac{4-5\alpha}{4}} + T_{ws}^{2\alpha}\kappa^{\frac{5\alpha-5}{2}} \right)\mathrm{poly}(\mathfrak{m}, \mathfrak{R}, \mathfrak{L}) \\
&\leq \frac{\epsilon_{ws}}{T_{ws}^{17/20}\kappa^{13/16}} + \frac{d\kappa^{-1/16}}{T_{ws}^{17/4}} + dT_{ws}^{3/20}\kappa^{-1/16}
\end{aligned}
$$

For our choice of $T, T_{ws}$, we have $\epsilon_{ws} \leq \frac{1}{\kappa}$. Overall the bound is

$$
\mathsf{FI}(\rho_\tau, \mu_0) \leq dT_{ws}^{3/20}\kappa^{-1/16}\mathrm{poly}(\mathfrak{m}, \mathfrak{L}, \mathfrak{R})
$$

$\square$

**Theorem 4.4.** *Suppose we run **Warm Start** phase with $T = \mathcal{O}\left(d\kappa \log(\kappa \mathsf{KL}(\gamma \| \mu_\infty))\right)$, $T_{ws} = \log \kappa d$, following which we run the **Annealing Phase** with $\delta = \kappa^{-1/4}$. This results in a $\tau = \kappa^{-3/16}$ satisfying*

$$
\mathsf{KL}(\mu_\tau \| \rho_{\tau\kappa/\delta}) \leq poly(d, 1/\kappa) \tag{3}
$$

*Proof.* From Lemma 4.3 using $T = \mathcal{O}(d^3\kappa \log(\kappa \mathsf{KL}(\gamma \| \mu_\infty)))$, we know that $\mathsf{KL}(\mu_{T_{ws}} \| \mathsf{Law}(X_{T_{ws}})) \leq \frac{1}{\kappa}$. Because $\lim_{t\to\infty} \mu_t$ is strongly log-concave, as shown in 4.3 for large $T_{ws}$ we can sample from $\mu_{T_{ws}}$ efficiently. From Lemma B.2 we have

$$\int_t^{T_{ws}} \|v_t\|_{L_2(\mu_t)}^2 dt \leq \int \frac{d^2 e^{-2t}}{(1 - e^{-2t})^8} \, \mathrm{poly}(\mathfrak{m}, \mathfrak{R}, \mathfrak{L}) \, dt$$

$$\leq \frac{d^2 e^{-2t} \, \mathrm{poly}(\mathfrak{m}, \mathfrak{R}, \mathfrak{L})}{(1 - e^{-2t})^8}$$

From here, we adapt the discretization analysis of Guo et al. (2024). We will repeat some of it below to highlight just the differences.

First note that $\nabla \log \mu_t$ inherits Lipschitzness from $\nabla \log p_t$ and $\nabla R$, following Lemma C.9:

$$\|\nabla \log \mu_t(x) - \nabla \log \mu_t(y)\| \leq \|\nabla \log p_t(x) - \nabla \log p_t(y) + \nabla R(y) - \nabla R(x)\|$$

$$\leq (1 + \mathfrak{L}e^{-t} + \mathfrak{R})\|x - y\|$$

By the corollary of Girsanov's Theorem referenced above, Lemma A.4, we see that

$$\mathsf{KL}(\mu_t \| \rho_t) = \mathsf{KL}(\mu_{T_{ws}} \| \mathsf{Law}(X_{T_{ws}})) + \frac{1}{4} \int_t^{T_{ws}} \mathbb{E}_{\{\mu_t\}} \| (\nabla \ln \mu_t(X_t) - \nabla \ln \mu_{k\delta}(X_{k\delta})) - v_t(X_t) \|^2 \, dt$$

$$\leq \mathsf{KL}(\mu_{T_{ws}} \| \mathsf{Law}(X_{T_{ws}})) + \int_t^{T_{ws}} \mathbb{E}_{\{\mu_t\}} \|\nabla \ln \mu_t(X_t) - \nabla \ln \mu_{k\delta}(X_{k\delta})\|^2 \, dt + \int_t^{T_{ws}} \mathbb{E}_{\{\mu_t\}} \|v_t(X_t)\|^2 \, dt$$

$$\leq \mathsf{KL}(\mu_{T_{ws}} \| \mathsf{Law}(X_{T_{ws}})) + \int_t^{T_{ws}} \mathrm{poly}(\mathfrak{R}, \mathfrak{L}) \mathbb{E}_{\{\mu_t\}} \|X_t - X_{k\delta}\|^2 \, dt + \int_t^{T_{ws}} \mathbb{E}_{\{\mu_t\}} \|v_t(X_t)\|^2 \, dt$$

We bound $X_t - X_{k\delta}$ by

$$\|X_t - X_{k\delta}\|^2 = \mathbb{E}_{\{\mu_t\}} \| \int_{k\delta}^t (\nabla \ln \mu_t + v_t)(X_t) \, dt + \sqrt{2(t - k\delta)}\eta \|^2, \qquad \eta \sim \gamma$$

$$\leq \int_{k\delta}^t \mathbb{E}_{\{\mu_t\}} \|\nabla \ln \mu_t\|^2 + \int_{k\delta}^t \mathbb{E}_{\{\mu_t\}} \|v_t(X_t)\|^2 \, dt + d\delta$$

We can bound $\mathbb{E}_{\{\mu_t\}} \|\nabla \ln \mu_t\|^2$.

$$\mathbb{E}_{\{\mu_t\}} \|\nabla \log \mu_t\|^2 \leq \mathbb{E}_{\mu_t} \|\nabla \log p_t + \nabla R\|^2$$

$$\leq \mathbb{E}_{\mu_t} \|\nabla \log p_t\|^2 + \mathbb{E}_{\mu_t} \|\nabla R\|^2 \leq \mathrm{poly}(\mathfrak{m}, \mathfrak{L}, \mathfrak{R}).$$

Putting these together, we have

$$\mathsf{KL}(\mu_t \| \rho_t) \leq \mathsf{KL}(\mu_{T_{ws}} \| \mathsf{Law}(X_{T_{ws}})) + (1 + \mathrm{poly}(\mathfrak{R}, \mathfrak{L})) \int_t^{T_{ws}} \mathbb{E}_{\{\mu_t\}} \|v_t(X_t)\|^2 \, dt + d\delta^2 \mathrm{poly}(\mathfrak{R}, \mathfrak{L})$$

$$+ \delta T_{ws} \, \mathrm{poly}(\mathfrak{R}, \mathfrak{L})$$

An important observation here is that because $v_t$ itself is a Wasserstein gradient, the quantity $\int_t^{T_{ws}} \mathbb{E}_{\{\mu_t\}} \|v_t(X_t)\|^2 \, dt$ depends inversely on the scale that we use for time. Suppose we reparameterize time to go from 0 to $T_{ws}\kappa$, rather than 0 to $T$. Let $\mathcal{A}_{t_1}^{t_2}$ denote the integral $\int_{t_1}^{t_2} \mathbb{E}_{\{\mu_t\}} \|v_t(X_t)\| \, dt$. Consider the change of variable $s = \kappa t$, so $s$ goes from 0 to $\kappa T$. Of course, we have the change of variables $ds = \kappa \, dt$, but also $v_s = \frac{1}{\kappa} v_t$. Then we have $\int_{t_1\kappa}^{t_2\kappa} \mathbb{E}_{\{\mu_s\}} \|v_s(X_s)\|^2 \, ds = \frac{1}{\kappa} \int_{t_1}^{t_2} \mathbb{E}_{\{\mu_t\}} \|v_t(X_t)\|^2 \, dt$. Over all, we have from lemma B.2

$$\mathsf{KL}(\mu_t \| \rho_{t\kappa/\delta}) \leq \mathsf{KL}(\mu_{T_{ws}} \| \mathsf{Law}(X_{T_{ws}})) + \frac{(1 + \delta \mathrm{poly}(\mathfrak{R}, \mathfrak{L}))}{\kappa} \int_t^{T_{ws}} \mathbb{E}_{\{\mu_t\}} \|v_t(X_t)\|^2 \, dt +$$

$$d\delta^2 \mathrm{poly}(\mathfrak{m}, \mathfrak{R}, \mathfrak{L}) + \delta \, \mathrm{poly}(\mathfrak{R}, \mathfrak{L})$$

$$\leq \mathsf{KL}(\mu_{T_{ws}} \| \mathsf{Law}(X_{T_{ws}})) + \frac{(1 + \delta \, \mathrm{poly}(\mathfrak{m}, \mathfrak{R}, \mathfrak{L}))}{T_{ws}\kappa} \frac{d^2}{(1 - e^{-2t})^3}$$

$$+ d\delta^2 \, \mathrm{poly}(\mathfrak{m}, \mathfrak{R}, \mathfrak{L}) + \delta \, \mathrm{poly}(\mathfrak{R}, \mathfrak{L})$$

$$\leq \mathsf{KL}(\mu_{T_{ws}} \| \mathsf{Law}(X_{T_{ws}})) + \frac{d^2(1 + \delta) \, \mathrm{poly}(\mathfrak{m}, \mathfrak{R}, \mathfrak{L})}{\kappa t^8} + \mathcal{O}\left(d\delta^2 + \delta\right)$$

We will take $i_{\mathsf{PS}} = (T_{ws}\kappa/\delta)^\alpha \delta/\kappa, \delta \asymp \kappa^{-1/4}$. Then we have

$$\mathsf{KL}\big(\mu_{i_{\mathsf{PS}}}\|\rho_{i_{\mathsf{PS}}\kappa/\delta}\big) \leq \mathsf{KL}\big(\mu_{T_{ws}}\|\mathsf{Law}(X_{T_{ws}})\big) + \frac{d^2}{T_{ws}^{8\alpha}\kappa^{10\alpha-9}} + \mathcal{O}(d\kappa^{-\frac{1}{2}} + \kappa^{-\frac{1}{4}})$$

Finally setting, $T = d^3\kappa^2\log\kappa\mathsf{KL}(\gamma\|\mu_\infty)$, $T_{ws} = \log\kappa d$, we have $\epsilon_{ws} = \frac{1}{\kappa}$ and choosing $\alpha = \frac{17}{20}$, we have

$$\mathsf{KL}\big(\mu_{i_{\mathsf{PS}}}\|\rho_{i_{\mathsf{PS}}\kappa/\delta}\big) \leq \mathcal{O}(d^2\kappa^{-1/2})$$

$\square$

[KL + FI] In algorithm 1, suppose we run **Warm Start** phase with $T = \mathcal{O}\big(d^3\kappa\log(\kappa\mathsf{KL}(\gamma\|\mu_\infty))\big)$, $T_{ws} = \log\kappa d$, following which we run the **Annealing Phase** with $\delta = \kappa^{1/4}$, then there is $\tau \leq \tilde{\mathcal{O}}(\kappa^{-3/16})$, such that $\rho_{\tau\kappa/\delta}$ simultaneously satisfies

- $\mathsf{KL}\big(\mu_\tau\|\rho_{\tau\kappa^{5/4}}\big) \leq \mathcal{O}(d\kappa^{-1/2})$, which implies $\mathsf{TV}\big(\rho_{\tau\kappa^{5/4}}, \mu_\tau\big) \leq \mathcal{O}(\sqrt{d\kappa^{-1/2}})$.

- $\mathsf{FI}\big(\rho_{\tau\kappa^{5/4}}, \mu_0\big) \leq \mathcal{O}(d\kappa^{-1/16})$

For this choice of $\kappa$, the algorithm has run time $\tilde{\mathcal{O}}(\kappa^{5/4})$.

*Proof.* All that is left to prove is that the run time is polynomial in $\kappa$. Note that we run the warm start phase for $\log\mathsf{KL}(\gamma\|\mu_\infty)/\epsilon$ iterations. Because $\gamma$ and $\mu_\infty$ are log-concave, we get $\mathsf{KL}(\gamma\|\mu_\infty) \leq \mathsf{LSI}(\mu_\infty)\mathsf{FI}(\gamma, \mu_\infty) = \mathcal{O}(d)$. The annealing phase lasts $T_{ws}\kappa/\delta = \mathcal{O}(\kappa^{5/4})$ time, since $T_{ws} = \mathcal{O}(\log d/\epsilon)$. $\square$

## C  MISCELLANEOUS BOUNDS

The role of this section is to establish bounds on various quantities. The main one is the global bound on $|\partial_t\log\mu_t|$ for $t > 0$, which we use in a couple of places.

- We use it to bound the Wasserstein derivative of the annealed path in Lemma B.2, and this is used with Girsanov's Theorem to bound the $\mathsf{KL}$ drift between the annealed LMC and the targets in Theorem 4.4.

- We also use it to bound the $\log\mu_t - \log\mu_\infty$ for large $t$ (Lemma C.7), which is used to show that we can transfer $\mathsf{FI}$ bounds from $\log\mu_t$ to $\log\mu_\infty$ in Theorems 4.3, 4.5.

We will begin with a statement about the sub-gaussianity of posteriors from sub-gaussian priors.

**Lemma C.1.** *Let $\mu$ denote the probability distribution of a sub-gaussian random variable with sub-gaussian parameter $\sigma$. Let $R \geq 0$ denote a smooth convex function with minima $\mathfrak{x}$ satisfying $R(\mathfrak{x}) = 0$ and $\nabla^2 R \preceq \mathfrak{R}I$. Let $\nu \propto \mu e^{-R}$ denote the posterior, and let $Y \sim \nu$. Then we have*

1. *$\nu$ is sub-gaussian with parameter $3\sigma(\sigma + \mathfrak{x}/2)\sqrt{\mathfrak{R}}$.*

2. *$\|\mathbb{E}_\nu Y\|^2 \leq 3\mathfrak{R}\sigma^2$.*

3. *$\mathbb{E}_\nu\|Y\|^2 \leq 9\mathfrak{R}\sigma^2(\sigma + \mathfrak{x}/2)^2 d + 3\mathfrak{R}\sigma^2$.*

*Proof.* 1. Let $X \sim \mu$. One of the characterizations of a $\sigma-$sub-gaussian random variable is decay of the tail probabilities $\Pr[X^\top\alpha > t] \leq 2e^{-\frac{t^2}{\sigma^2}}$. Let $Y \sim \nu$. We have

$$\Pr[Y^\top\alpha > t] = \int_t^\infty \frac{\int_{x^\top\alpha=s}\mu(x)e^{-R(x)}}{\int\mu(x)e^{-R(x)}\,dx}\,ds.$$

The partition function can be lower bounded as

$$
\int \mu(x) e^{-R(x)} \, dx \geq \int_{\|x\| < 2\mathfrak{m}+\mathfrak{x}} \mu(x) e^{-R(x)} \, dx
$$

$$
\geq \left( \min_{\|x\| \leq 2\mathfrak{m}+\mathfrak{x}} e^{-R(x)} \right) \int_{\|x\| < 2\mathfrak{m}+\mathfrak{x}} \mu(x) \, dx
$$

$$
= e^{-\max_{\|x\| \leq 2\mathfrak{m}+\mathfrak{x}} R(x)} \Pr[X < 2\mathfrak{m} + \mathfrak{x}] \geq e^{-2(\mathfrak{m}+\mathfrak{x}/2)^2 \mathfrak{R}}/2
$$

The tail can now be upper bounded as

$$
\Pr[Y^\top \alpha > t] \leq \int_t^\infty \frac{\int_{x^\top \alpha = s} \mu(x) e^{-R(x)}}{\int \mu(x) e^{-R(x)} \, dx} \, ds
$$

$$
\leq 2 e^{2(\mathfrak{m}+\mathfrak{x}/2)^2 \mathfrak{R}} \int_t^\infty \int_{x^\top \alpha = s} \mu(x) \, ds
$$

$$
\leq 2 e^{2(\mathfrak{m}+\mathfrak{x}/2)^2 \mathfrak{R}} \Pr[X^\top \alpha > t] \leq 4 e^{2(\mathfrak{m}+\mathfrak{x}/2)^2 \mathfrak{R} - \frac{t^2}{\mathfrak{m}^2}}.
$$

Of course this bound is vacuous until $4 e^{2(\mathfrak{m}+\mathfrak{x}/2)^2 \mathfrak{R} - \frac{t^2}{\mathfrak{m}^2}} < 1$, which happens when

$$
2(\mathfrak{m} + \mathfrak{x}/2)^2 \mathfrak{R} - \frac{t^2}{\mathfrak{m}^2} < -\log 4 \implies t > \sqrt{\mathfrak{m}^2((\mathfrak{m} + \mathfrak{x}/2)^2 \mathfrak{R} + \log 4)}.
$$

When $t > \sqrt{\mathfrak{m}^2((\mathfrak{m} + \mathfrak{x}/2)^2 \mathfrak{R} + \log 4)}$, we have $2(\mathfrak{m} + \mathfrak{x}/2)^2 \mathfrak{R} - \frac{t^2}{\mathfrak{m}^2} < -\frac{t^2}{\mathfrak{m}^2(2(\mathfrak{m}+\mathfrak{x}/2)^2\mathfrak{R}+2)}$. Overall, this shows that $\nu$ is a sub-gaussian distribution with parameter $\mathfrak{m}\sqrt{2(\mathfrak{m} + \mathfrak{x}/2)^2 \mathfrak{R} + 2} \leq 3\mathfrak{m}(\mathfrak{m} + \mathfrak{x}/2)\sqrt{\mathfrak{R}}$.

2. From Donsker-Varadhan, we have $\mathbb{E}_{\mu_t} X \leq \mathsf{KL}(\mu_t \| p_t) + \log \mathbb{E}_{p_t} e^X$. From sub-gaussianity we have $\log E_{p_t} e^X \leq e^{\mathfrak{m}^2/2}$. The $\mathsf{KL}$ can be bounded as

$$
\mathsf{KL}(\mu_t \| p_t) = -\mathbb{E}_{\mu_t} R - \log \mathbb{E}_{p_t} e^{-R}
$$

$$
\leq -\log \mathbb{E}_{p_t} e^{-R} \qquad\qquad \cdots R > 0
$$

$$
= -\log \int e^{-R(x)} p_t(x) \, dx
$$

$$
\leq -\log \int_{\|x\| \leq \mathfrak{m}_2} e^{-R(x)} p_t(x) \, dx
$$

$$
\leq -\log e^{-R_1 \mathfrak{m}^2}(1 - 2e^{-1})
$$

$$
\leq 2 + \mathfrak{R}\mathfrak{m}^2 \leq 3\mathfrak{R}\mathfrak{m}^2
$$

Here the last inequality follows because $R(x) \leq \mathfrak{m}^2 \mathfrak{R}$ in the region $\|x\| \leq \mathfrak{m}_2$, and $\Pr_{p_t}(X > \mathfrak{m}_2) \leq 2e^{-1}$ from sub-gaussianity.

3. For simplicity we will consider the zero-mean case, the general, full second moment will be the sum of the centered second moment and the square of the mean. We have $\mathrm{Var}(Y^\top \alpha) \leq 9\mathfrak{R}\sigma^2(\sigma + \mathfrak{x}/2)^2$ for all $\alpha$. Now consider an orthonormal basis $\{\alpha_i\}$, summing the above relation for each of them we have

$$
\sum_i \mathrm{Var}(Y^\top \alpha_i) = \sum_i \mathbb{E}_\nu (Y^\top \alpha_i)^2 = \mathbb{E}_\nu \sum_i (Y^\top \alpha_i)^2
$$

$$
= \mathbb{E}_\nu \sum_i (Y^\top \alpha_i \alpha_i^\top Y) = \mathbb{E}_\nu \sum_i (Y^\top \alpha_i \alpha_i^\top Y)
$$

$$
= \mathbb{E}_\nu (Y^\top \left( \sum_i \alpha_i \alpha_i^\top \right) Y) = \mathbb{E}_\nu \|Y\|^2
$$

Finally, if $\mathbb{E}_\nu Y \neq 0$, we write

$$
\mathbb{E}_\nu \|Y\|^2 = \mathbb{E}_\nu \|Y - \mathbb{E}_\nu Y\|^2 + \|\mathbb{E}_\nu Y\|^2 = 9\mathfrak{R}\sigma^2(\sigma + \mathfrak{x}/2)^2 d + 3\mathfrak{R}\sigma^2.
$$

$\square$

**Lemma C.2.** *Let $p_0$ by $\mathfrak{m}$-subgaussian. The law of the OU process $p_t$ is subgaussian with norm $\mathfrak{m}e^{-t} + (1 - e^{-2t})$.*

We also need the following, about moments of subgaussian random variables.

**Lemma C.3.** *Let $\nu$ denote a $\mathfrak{m}-$subgaussian distribution. For any $f$ satisfying $f(x) \leq \sum_{i=1}^{k} a_i \|x\|^k$, we have*

$$\mathbb{E}_\nu f(x) \leq \sum_{i=1}^{k} (2\mathfrak{m})^i i^{i/2} a_i.$$

*Proof.* Follows from standard results of subgaussian random variables. $\square$

**Lemma C.4.** *The density $p_t$ is upper bounded by*

$$p_t \leq \frac{1}{(2\pi(1 - e^{-2t}))^{d/2}}$$

*Proof.* We have

$$p_t(x) = \int p(e^t y)\gamma_{1-e^{-2t}}(x - y)dy \leq \sup_y \gamma_{1-e^{-2t}}(y) \int p(e^t y)dy = \frac{1}{(2\pi(1 - e^{-2t}))^{d/2}}$$

$\square$

**Note:** Of course, the density can blow up at $t = 0$ (that is, for unsmoothed distributions), but once we add heat the density is bounded.

**Lemma C.5.** *Let $p_t$ denote the law of $X_t$, where $X_0 \sim p_0$ and $X_t$ satisfies OU. Then we have*

$$|\partial_t \log p_t| \leq \frac{e^{-t}}{(1 - e^{-2t})^4} \sum_{i=0}^{2} a_i \|x\|^i.$$

*For $a_i = poly(\mathfrak{m}, \mathfrak{R}, \mathfrak{L})$.*

*Proof.* We will directly compute $\partial_t \log p_t$

$$\partial_t \log p_t = \partial_t \log p_t = \frac{\partial_t p_t}{p_t} \qquad\qquad \text{Lemma } A.2(5)$$

$$= \frac{-\nabla \cdot (p_t \nabla \log \frac{p_t}{\gamma})}{p_t} \qquad\qquad \text{Fokker-Planck}$$

$$= \frac{-\nabla p_t \cdot \nabla \log \frac{p_t}{\gamma} - p_t \Delta \log \frac{p_t}{\gamma}}{p_t} \qquad\qquad \text{Lemma } A.2(4)$$

$$= -\nabla \log p_t \cdot \nabla \log \frac{p_t}{\gamma} - \Delta \log \frac{p_t}{\gamma} \qquad\qquad \text{Lemma } A.2(5)$$

$$= \nabla \log p_t \cdot \nabla \log \gamma - \left(\Delta \log \frac{p_t}{\gamma} + \|\nabla \log p_t\|^2\right)$$

We have

$$\Delta \log \frac{p_t}{\gamma} = \Delta \log p_t - \Delta \log \gamma = d + \nabla \cdot \left( \frac{\nabla p_t}{p_t} \right) \qquad \text{Lemma } A.3(3)$$

$$= d - \frac{\|\nabla p_t\|^2}{p_t^2} + \frac{\Delta p_t}{p_t} \qquad \text{Lemma } A.2(4)$$

$$= d + \frac{(p \circ e^t) * \Delta \gamma_{1-e^{-2t}}}{(p \circ e^t) * \gamma_{1-e^{-2t}}} - \|\nabla \log p_t\|^2 \qquad \text{Lemma } A.2(3,5)$$

$$= d + \frac{\int (p \circ e^t)(x-y) \left( \frac{\|y\|^2}{(1-e^{-2t})^2} - \frac{d}{1-e^{-2t}} \right) \gamma_{1-e^{-2t}}(y) \, dy}{\int (p \circ e^t)(x-y) \gamma_{1-e^{-2t}}(y) \, dy} - \|\nabla \log p_t\|^2 \qquad \text{Lemma } A.3(2)$$

$$= \frac{e^{-2t}}{e^{-2t}-1} d + \frac{\int (p \circ e^t)(x-y) \frac{\|y\|^2}{(1-e^{-2t})^2} \gamma_{1-e^{-2t}}(y) \, dy}{\int (p \circ e^t)(x-y) \gamma_{1-e^{-2t}}(y) \, dy} - \|\nabla \log p_t\|^2$$

Note that $\circ$ refers to composition. As a shorthand, we will write $c_x(y) = \frac{(p \circ e^t)(x-y) \gamma_{1-e^{-2t}}(y)}{\int (p \circ e^t)(x-y) \gamma_{1-e^{-2t}}(y) \, dy}$.
Note that $c_x(y)$ can be interpreted as a posterior. Let $\tau_x$ denote the isometry $\tau_x(y) = x - y$, then we can interpret $\frac{1}{e^{dt}} p \circ e^t \circ \tau_x$ as a prior, and $\gamma$ is a likelihood. At this stage, the following identity about the gradient will be useful

$$\nabla \log p_t = \frac{\nabla p_t}{p_t} \qquad \text{Lemma } A.2(5)$$

$$= \frac{(p \circ e^t) * \nabla \gamma_{1-e^{-2t}}}{(p \circ e^t) * \gamma_{1-e^{-2t}}} \qquad \text{Lemma } A.2(2)$$

$$= \frac{(p \circ e^t) * \frac{y}{1-e^{-2t}} \gamma_{1-e^{-2t}}}{(p \circ e^t) * \gamma_{1-e^{-2t}}} \qquad \text{Lemma } A.3(1)$$

$$= \frac{\int (p(e^t(x-y)) \frac{y}{1-e^{-2t}} \gamma_{1-e^{-2t}}(y) \, dy}{\int p(e^t(x-y)) \gamma_{1-e^{-2t}}(y) \, dy}$$

$$= \frac{1}{1-e^{-2t}} \int y \, c_x(y) \, dy \qquad (6)$$

We have

$$\Delta \log \frac{p_t}{\gamma} + \|\nabla \log p_t\|^2 - \nabla \log p_t \cdot \nabla \log \gamma$$

$$= \frac{e^{-2t}}{e^{-2t}-1} d + \frac{1}{(1-e^{-2t})^2} \int \|y\|^2 c_x(y) \, dy - \nabla \log p_t \cdot \nabla \log \gamma$$

$$= \frac{e^{-2t}}{e^{-2t}-1} d + \int \left( \frac{\|y\|^2}{(1-e^{-2t})^2} - \frac{y \cdot x}{1-e^{-2t}} \right) c_x(y) \, dy$$

$$= \frac{e^{-2t}}{e^{-2t}-1} d + \frac{1}{(1-e^{-2t})^2} \int \left( \|x-y\|^2 - x \cdot (y-x) + e^{-2t} y \cdot x \right) c_x(y) \, dy$$

Lets consider the terms in the integral.

$$\int \|x-y\|^2 c_x(y) \, dy$$

$$\leq \int \left( \|\mathbb{E}_{y \sim c_x(\cdot)} y - y\|^2 + \|x - \mathbb{E}_{y \sim c_x(\cdot)} y\|^2 \right) c_x(y) \, dy$$

$$= \int \|\mathbb{E}_{y \sim c_x(\cdot)} y - y\|^2 c_x(y) \, dy + \|x - \mathbb{E}_{y \sim c_x(\cdot)} y\|^2$$

We will now use Lemma C.1 to bound these terms.

The first is just the variance of the posterior $c_x$. Note that in the application of the lemma, the prior is $p_t \circ e^t \circ \tau_x$, which has mean $x$ (since $p_t$ has zero mean) and subgaussian parameter $\mathfrak{m} e^{-t}$, and the

likelihood is $\gamma_{1-e^{-2t}}$, which has minima at $\mathfrak{x} = x$, and Hessian bounded by $\mathfrak{R} = \frac{1}{1-e^{-2t}}$. By Lemma C.1 (3) we have

$$\int \|\mathbb{E}_{y\sim c_x(\cdot)} y - y\|^2 c_x(y) \, dy \leq \frac{9}{1-e^{-2t}} e^{-2t} \mathfrak{m}^2 (\mathfrak{m} e^{-t} + \frac{\|x\|}{2})^2 d + \frac{3}{1-e^{-2t}} \mathfrak{m}^2 e^{-2t}.$$

The second is controlled by Lemma C.1 (2), since $\mathbb{E}_{X\sim p_t\circ e^t\circ \tau_x} X = x$. We have that

$$\|x - \mathbb{E}_{Y\sim c_x} Y\|^2 \leq 9\mathfrak{m}^4 \frac{e^{-4t}}{(1-e^{-2t})^2}.$$

For readability we will assume $\mathfrak{m}, d > 1$. Then we have

$$\int \|x - y\|^2 c_x(y) \, dy \leq \frac{1}{(1-e^{-2t})^2} 9\mathfrak{m}^2 d e^{-2t} \left(3\mathfrak{m}^2 + \|x\|^2\right).$$

Similarly

$$\int \|x - y\| c_x(y) \, dy \leq \left(\int \|x - y\|^2 c_x(y) \, dy\right)^{1/2}$$

$$\leq \frac{1}{1-e^{-2t}} 3\mathfrak{m} e^{-t} \sqrt{d\left(3\mathfrak{m}^2 + \|x\|^2\right)}$$

So we have

$$\left| \Delta \log \frac{p_t}{\gamma} + \|\nabla \log p_t\|^2 - \nabla \log p_t \cdot \nabla \log \gamma \right|$$

$$= \left| \frac{e^{-2t}}{e^{-2t}-1} d + \frac{1}{(1-e^{-2t})^2} \int \left( \|x-y\|^2 - x\cdot(y-x) + e^{-2t} y\cdot x \right) c_x(y) \, dy \right|$$

$$\leq \frac{e^{-2t}}{1-e^{-2t}} d + \frac{1}{(1-e^{-2t})^4} \left| 12\mathfrak{m}^2 d e^{-t} \left(3\mathfrak{m}^2 + \|x\|^2\right) + \int e^{-2t} y\cdot x c_x(y) \, dy \right|$$

$$\leq \frac{e^{-2t}}{1-e^{-2t}} d + \frac{12\mathfrak{m}^2 d e^{-t} \left(3\mathfrak{m}^2 + \|x\|^2\right)}{(1-e^{-2t})^4} + \left| \frac{e^{-2t}}{(1-e^{-2t})} \nabla \log p_t \cdot x \right| \qquad \text{from Equation (6)}$$

$$\leq \frac{12\mathfrak{m}^2 d e^{-t} \left(3\mathfrak{m}^2 + \|x\|^2\right)}{(1-e^{-2t})^4} + \frac{e^{-2t}}{(1-e^{-2t})} (d + \mathfrak{L}\|x\| + \mathfrak{L}\|x\|^2)$$

We can write this as $|\partial_t \log p_t| \leq \frac{e^{-t}}{(1-e^{-2t})^4} \sum_{i=0}^2 a_i \|x\|^i$ for $a_i = d\text{poly}(\mathfrak{m}, \mathfrak{L}, \mathfrak{R})$. $\qquad \square$

**Lemma C.6.** *We have $|\partial_t \log \mu_t| \leq \frac{e^{-t}}{(1-e^{2t})^4} \sum_{i=0}^2 a_i \|x\|^i$ for $a_i = d\text{poly}(\mathfrak{m}_2, \mathfrak{L}, \mathfrak{R})$.*

*Proof.* We have

$$\partial_t \log \mu_t = \partial_t \log \frac{p_t e^{-R}}{\int p_t e^{-R}} = \partial_t \log p_t - \partial_t R - \partial_t \log \int p_t e^{-R}$$

$$= \partial_t \log p_t - \frac{\partial_t \int p_t e^{-R}}{\int p_t e^{-R}} = \partial_t \log p_t - \frac{\int p_t \partial_t \log p_t e^{-R}}{\int p_t e^{-R}}$$

$$\leq \partial_t \log p_t + \mathbb{E}_{\mu_t} \partial_t \log p_t \leq \partial_t \log p_t + \mathbb{E}_{\mu_t} |\partial_t \log p_t|$$

$$\leq \frac{e^{-t}}{(1-e^{2t})^4} \sum_{i=0}^2 a_i \|x\|^i \qquad\qquad C.5, C.3.$$

For $a_i = d\text{poly}(\mathfrak{m}_2, \mathfrak{L}, \mathfrak{R})$ $\qquad \square$

**Lemma C.7.** *Let $\mu_t \propto p_t e^{-R}$. For $T > 1$, we have*

$$|\log \mu_T(x) - \log \mu_\infty(x)| \leq \frac{e^{-T}}{(1-e^{-2T})^4} \sum_{i=0}^2 a_i \|x\|^i.$$

*Where $a_i = \text{poly}(\mathfrak{m}, \mathfrak{L}, \mathfrak{R}, d)$.*

*Proof.*

$$\left|\log \mu_T - \log \mu_\infty\right| = \left|\int_T^\infty \partial_t \log \mu_t \; dt\right| \leq \int_T^\infty |\partial_t \log \mu_t| \; dt$$

$$\leq \int_T^\infty \frac{e^{-t}}{(1-e^{-2t})^4} \sum_{i=0}^2 a_i \|x\|^i \; dt$$

$$\leq \frac{e^{-T}}{(1-e^{-2T})^4} \sum_{i=0}^2 a_i \|x\|^i$$

$\square$

**Lemma C.8.** *Let $p_{t\to 0}(x|x_t) = \Pr\{e^{-t}x + \sqrt{1-e^{-2t}}\eta = x_t, \; \eta \sim \gamma\}$ be the posterior of the OU process conditioned on a future iterate. We have*

$$\nabla \log p_t(x) = \mathbb{E}_{X \sim p_{t\to 0}(\cdot|x)} \nabla \log p_0(X)$$

*Proof.* Please see Proposition 2.1 of (Bortoli et al., 2024). $\square$

**Lemma C.9.** *Let $X_0 \sim p_0$ with $\nabla \log p_0$ being $\mathfrak{L}-$Lipshitz for $\mathfrak{L} > 1$, and let $X_t$ denote the OU process run for time $t$, with law $X_t \sim p_t$. Then $\nabla \log p_t$ is $\mathfrak{L}$-Lipshitz.*

# D  FI IS NOT SUFFICIENT

**Lemma D.1.** *Take two distributions $\gamma_1, \gamma_2$. Let $\gamma_{1|\mathcal{B}_\varepsilon(x)}$ (respectively, $\gamma_{2|\mathcal{B}_\varepsilon(x)}$) denote the distribution $\gamma_1$ conditioned on being within a ball of radius $\epsilon$ around the point $x$. Then we have*

$$\mathbb{E}_{X \sim \gamma_1} \mathsf{KL}\big(\gamma_{1|\mathcal{B}_\varepsilon(X)} \| \gamma_{2|\mathcal{B}_\varepsilon(X)}\big) \lesssim \varepsilon \mathsf{FI}(\gamma_1, \gamma_2).$$

*Proof.* For $\gamma$ smooth around $x$, $\gamma(y) = \gamma(x) + (y-x)^\top \gamma(x) + \mathcal{O}(\|y-x\|^2)$, so

$$\int_{y \in \mathcal{B}_\epsilon(x)} \gamma(y) \; dy = \big(\gamma(x) + O(\epsilon^2)\big) \operatorname{vol}(\mathcal{B}_\epsilon(x))$$

and

$$\gamma_{|\mathcal{B}_\varepsilon(x)}(x) = \frac{\gamma(x)}{\int_{y \in \mathcal{B}_\varepsilon(x)} \gamma(y) \; dy} = \frac{\gamma(x)}{(\gamma(x) + \Theta(\epsilon^2)) \operatorname{vol}(\mathcal{B}_\epsilon)} =_\epsilon \frac{1}{\operatorname{vol}(\mathcal{B}_\epsilon)}.$$

Let $\nabla_a f \mid_b$ denote the gradient with respect to $a$ evaluated at $b$, then we also have

$$\nabla_z \log \gamma_{|\mathcal{B}_\varepsilon(x)}(z) \mid_{z=x} = \nabla_z \log \gamma(z) \mid_{z=x} - \nabla_z \log \int_{y \in \mathcal{B}_\varepsilon(x)} \gamma(y) \; dy \mid_{z=x} = \nabla \log \gamma(x),$$

so we have:

$$\mathbb{E}_{X \sim \gamma_1} \mathsf{KL}\big(\gamma_{1|\mathcal{B}_\varepsilon(X)} \| \gamma_{2|\mathcal{B}_\varepsilon(X)}\big)$$

$$= \mathbb{E}_{X \sim \gamma_1} \mathbb{E}_{Y \sim \gamma_{1|\mathcal{B}_\varepsilon(X)}} [\log \gamma_{1|\mathcal{B}_\varepsilon(X)}(Y) - \log \gamma_{2|\mathcal{B}_\varepsilon(X)}(Y)]$$

$$= \mathbb{E}_{X \sim \gamma_1} \mathbb{E}_{Y \sim \gamma_{1|\mathcal{B}_\varepsilon(X)}} [\log \gamma_{1|\mathcal{B}_\varepsilon(X)}(X + (Y-X)) - \log \gamma_{2|\mathcal{B}_\varepsilon(X)}(X + (Y-X))]$$

$$= \mathbb{E}_{X \sim \gamma_1} \mathbb{E}_{Y \sim \gamma_{1|\mathcal{B}_\varepsilon(X)}} [\log \gamma_{1|\mathcal{B}_\varepsilon(X)}(X) - \log \gamma_{2|\mathcal{B}_\varepsilon(X)}(X) + (Y-X)^\top (\nabla \log \gamma_{1|\mathcal{B}_\varepsilon(X)}(X) - \nabla \log \gamma_{2|\mathcal{B}_\varepsilon(X)}(X))]$$

$$\approx \mathbb{E}_{X \sim \gamma_1} \mathbb{E}_{Y \sim \gamma_{1|\mathcal{B}_\varepsilon(X)}} [(Y-X)^\top (\nabla \log \gamma_{1|\mathcal{B}_\varepsilon(X)}(Y) - \nabla \log \gamma_{2|\mathcal{B}_\varepsilon(X)}(Y))]$$

$$\leq \varepsilon \mathbb{E}_{X \sim \gamma_1} \mathbb{E}_{Y \sim \gamma_{1|\mathcal{B}_\varepsilon(X)}} [\|\nabla \log \gamma_{1|\mathcal{B}_\varepsilon(X)}(X) - \nabla \log \gamma_{2|\mathcal{B}_\varepsilon(X)}(X)\|]$$

$$= \varepsilon \; \mathbb{E}_{X \sim \gamma_1} [\|\nabla \log \gamma_1(X) - \nabla \log \gamma_2(X)\|] = \varepsilon \mathsf{FI}(\gamma_1, \gamma_2)$$

$\square$

**Lemma D.2.** *Let*

$$p_0 = \frac{1}{2}\mathcal{N}(\mathbf{0}, I) + \frac{1}{2}\mathcal{N}\left(\lambda\begin{bmatrix}1\\1\end{bmatrix}, I_2\right), \qquad R(\mathbf{x}) = \frac{1}{2\eta}\|diag([0,1])\mathbf{x}\|^2$$

*Let* $\frac{1}{\eta'} = 1 + \frac{1}{\eta}$, *and let* $A_\square = diag([1, \square])$ *for any* $\square$. *Then the posterior can be written as*

$$\mu_0 = \alpha_0\mathcal{N}\left(\mathbf{0}, A_{\eta'}\right) + (1 - \alpha_0)\mathcal{N}\left(\lambda A_{\eta'}\begin{bmatrix}1\\1\end{bmatrix}, A_{\eta'}\right)$$

*with* $\alpha_0 = \frac{1}{1+e^{-\frac{\lambda^2}{1+\eta}}}$, *and the distribution*

$$\mu_0' = \frac{1}{2}\mathcal{N}\left(\mathbf{0}, A_{\eta'}\right) + \frac{1}{2}\mathcal{N}\left(\lambda A_{\eta'}\begin{bmatrix}1\\1\end{bmatrix}, A_{\eta'}\right)$$

*satisfies*

$$FI(\mu_1, \mu_2) \le \lambda e^{2\lambda^2/(1+\eta)-\lambda^2/8}$$

*Proof.* Take the marginals of $\mu_0, \mu_0'$ onto the two coordinates (denoted "$x$" and "$y$").

$$\mu_{0,x}' = \frac{1}{2}\mathcal{N}(0,1) + \frac{1}{2}\mathcal{N}(\lambda, 1) \qquad \mu_{0,x} = \alpha_0\mathcal{N}(0,1) + (1-\alpha_0)\mathcal{N}(\lambda, 1)$$

$$\mu_{0,y}' = \frac{1}{2}\mathcal{N}(0, \eta') + \frac{1}{2}\mathcal{N}(\frac{\lambda\eta}{1+\eta}, \eta') \qquad \mu_{0,x} = \alpha_0\mathcal{N}(0, \eta') + (1-\alpha_0)\mathcal{N}(\frac{\lambda\eta}{1+\eta}, \eta')$$

We have $FI(\mu_0', \mu_0) \le FI(\mu_{0,x}', \mu_{0,x}) + FI(\mu_{0,y}', \mu_{0,y})$. We can apply Lemma D.3 to each of these marginals seperately to get

$$FI(\mu_0', \mu_0) \le \frac{\lambda}{(1-\alpha_0)^2}e^{-\lambda^2/8} \le \lambda e^{2\lambda^2/(1+\eta)-\lambda^2/8}.$$

$\square$

**Lemma D.3.** *Consider two mixtures of scalar Gaussians*

$$\mu_1 = \alpha_1\mathcal{N}(0, \sigma) + (1-\alpha_1)\mathcal{N}(\beta, \sigma)$$
$$\mu_2 = \alpha_2\mathcal{N}(0, \sigma) + (1-\alpha_2)\mathcal{N}(\beta, \sigma)$$

*with* $\alpha_2 > \alpha_1 > \frac{1}{2}$. *We have*

$$FI(\mu_1, \mu_2) \le \frac{(1-\alpha_1)^2}{(1-\alpha_2)^2}\frac{\beta}{\sigma}e^{-\beta^2/8\sigma^2}.$$

*Proof.* For convenience, we write $\gamma_1 = \mathcal{N}(0, \sigma), \pi_2 = \mathcal{N}(\beta, \sigma)$. Note that $\nabla\log\pi_1 = -x/\sigma, \nabla\log\pi_2 = -(x-\beta)/\sigma$. We upper bound the $FI(\mu_1, \mu_2)$ as follows (this follows the argument in Balasubramanian et al. (2022) very closely, just with modified parameters)

$$\nabla\log^{\mu_1/\mu_2} = \frac{1}{\mu_1\mu_2}\left(\mu_2\left(\alpha_1\nabla\pi_1 + (1-\alpha_1)\nabla\pi_2\right) - \mu_1\left(\alpha_2\nabla\pi_1 + (1-\alpha_2)\nabla\pi_2\right)\right)$$

$$= \frac{(\alpha_2-\alpha_1)}{\mu_1\mu_2}\left(\pi_1\nabla\pi_2 - \pi_2\nabla\pi_1\right)$$

$$= (\alpha_2-\alpha_1)\frac{\pi_1\pi_2}{\mu_1\mu_2}\left(\nabla\log\pi_2 - \nabla\log\pi_1\right) = (\alpha_2-\alpha_1)\frac{\pi_1\pi_2}{\mu_1\mu_2}\frac{\beta}{\sigma}$$

so we have

$$\mathsf{FI}(\mu_1, \mu_2) = \mathbb{E}[(\nabla \log \mu_1/\mu_2)^2]$$

$$= (\alpha_2 - \alpha_1)^2 \frac{\beta^2}{\sigma^2} \int \frac{\pi_1^2 \pi_2^2}{\mu_1^2 \mu_2^2} \, d\mu_1$$

$$= (\alpha_2 - \alpha_1)^2 \frac{\beta^2}{\sigma^2} \int \frac{\pi_1^2 \pi_2^2}{\mu_1 \mu_2^2} \, dx$$

$$= (\alpha_2 - \alpha_1)^2 \frac{\beta^2}{\sigma^2} \int \frac{\pi_1^2 \pi_2^2}{(\alpha_1 \pi_1 + (1 - \alpha_1)\pi_2)(\alpha_2 \pi_1 + (1 - \alpha_2)\pi_2)^2} \, dx$$

$$\leq (\alpha_2 - \alpha_1)^2 \frac{\beta^2}{\sigma^2} \left( \frac{1}{(1-\alpha_1)\alpha_2(1-\alpha_2)} \int_{x \leq \beta/2} \frac{\pi_2^2}{\pi_1} \, dx + \frac{1}{(1-\alpha_1)(1-\alpha_2)^2} \int_{x \geq \beta/2} \frac{\pi_1^2}{\pi_2} \, dx \right)$$

$$\leq \frac{(\alpha_2 - \alpha_1)^2}{(1-\alpha_1)(1-\alpha_2)^2} \frac{\beta^2}{\sigma^2} \left( \int_{x \leq \beta/2} \frac{\pi_2^2}{\pi_1} \, dx + \int_{x \geq \beta/2} \frac{\pi_1^2}{\pi_2} \, dx \right)$$

Finally

$$\int_{x \leq \beta/2} \pi_2^2/\pi_1 = \frac{1}{\sqrt{2\pi}\sigma} \int_{x \leq \beta/2} e^{-(x-\beta)^2 + \frac{1}{2}x^2} = \frac{e^{\beta^2}}{\sqrt{2\pi}\sigma} \int_{x \leq \beta/2} e^{-\frac{1}{2}(x-2\beta)^2} \leq \frac{1}{\sqrt{2\pi}\sigma\beta} e^{-9\beta^2/8}.$$

This also holds for the other term $\int_{x \geq \beta/2} \frac{\pi_1^2}{\pi_2}$. Overall we have $\mathsf{FI}(\mu_1, \mu_2) \leq \frac{(\alpha_2 - \alpha_1)^2}{(1-\alpha_1)(1-\alpha_2)^2} \frac{\beta}{\sigma} e^{-\beta^2/8\sigma^2}.$

$\square$

# E  SYNTHETIC SIMULATIONS

We have include some synthetic simulations of our method below. We use three priors for illustration (Figure 6), a mixture of gaussians with 5 components, a set of vertical bars, some of which have gaps in them (similar to the discussion in Remark 4.7), and a pair of "moons". We illustrate the posterior sampling algorithm with two choices of measurement models, $y = Ax + \eta$ for $A = \begin{pmatrix} 1 & 0 \\ 0 & 0 \end{pmatrix}$ (Figure 7) and also simply $y = x + \eta$ for $\eta \sim \mathcal{N}(0, \frac{1}{\Re})$ (Figure 9). The sampler of Algorithm 1, with $\kappa = 400$ and $T_{ws}/\delta = 200$ total noising levels is shown in Figures 8 and 10.

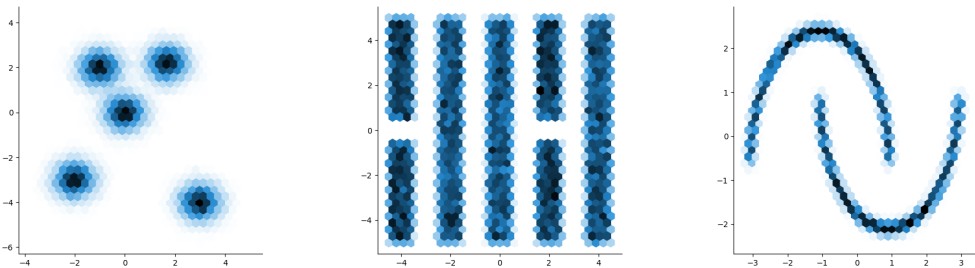

Figure 6: Three priors used in our experiments. A Mixture-of-Gaussians prior on the left, a "Vertical Bars" prior in the center (similar to Remark 4.7), and a "moons" prior on the right.

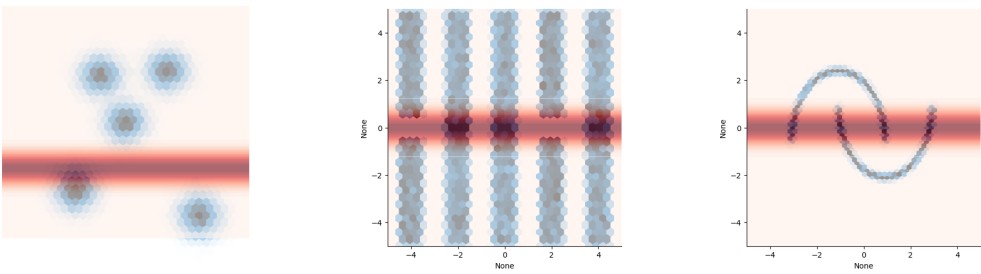

Figure 7: Likelihood functions used to define the posterior. $R(x) = \Re\|Ax\|^2$ where $A = \begin{pmatrix} 1 & 0 \\ 0 & 0 \end{pmatrix}$. Essentially these are "noisy projections", somewhat analogous to an inpainting problem (one coordinate is seen, the other is not).

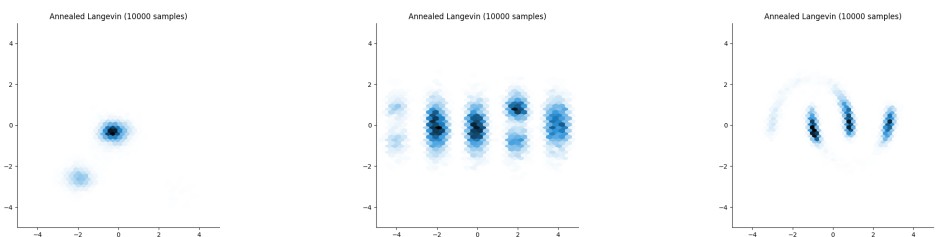

Figure 8: Resulting sampler, run with $\kappa = 400$. Shown are hex-jointplots of 10000 samples each.

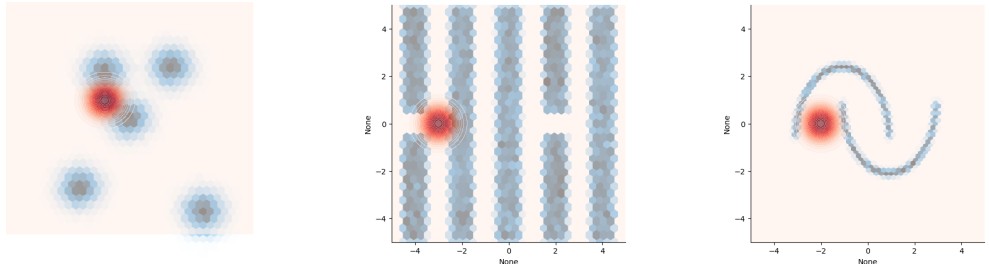

Figure 9: Likelihood functions used to define the posterior, corresponding to a noisy gaussian measurement $R(x) = \mathfrak{R}\|x\|^2$.

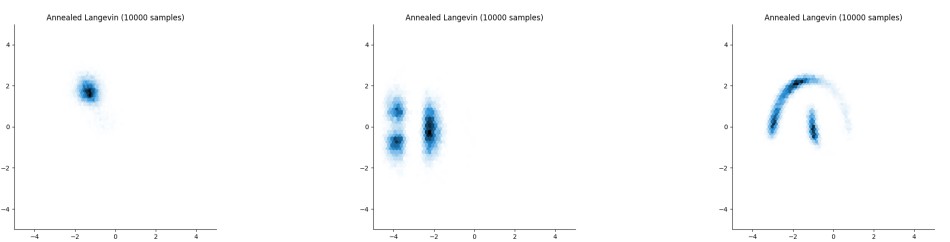

Figure 10: Resulting sampler, run with $\kappa = 400$, with 200 levels of noising (so a total of 80000 iterations). Shown are hex-jointplots of 10000 samples each.

We see that each of the modes are discovered (avoiding the mode collapse phenomemon associated with FI).

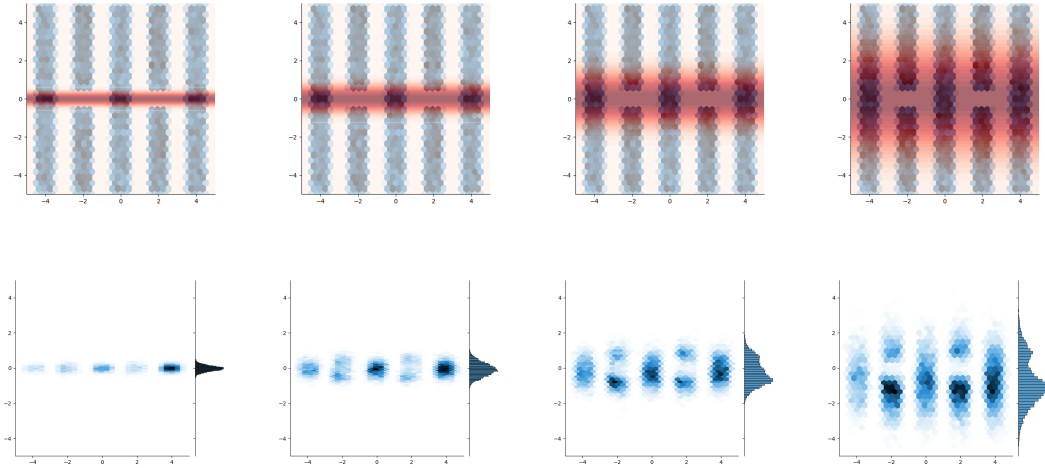

Figure 11: Here we demonstrate the consequences of changing the variance of the noise used in the measurement (which is related to $\mathfrak{R}$ as we see in 4.2). For large values of $\mathfrak{R}$, the gap in the second and fourth vertical bars is much less stark, but the mass dedicated to these bars does not vanish.

