# OpenReview forum: "Efficient Approximate Posterior Sampling with Annealed Langevin Monte Carlo"
_ICLR.cc/2026/Conference — ICLR 2026 Poster_

### Official Review · Reviewer_qQ6U · 2025-10-28

**Soundness:** 2
**Presentation:** 2
**Contribution:** 3
**Rating:** 4
**Confidence:** 3

**Summary:**

The paper introduces an algorithm for posterior sampling under oracle access to the score function of the prior distribution along the OU process. Their key insight is using annealed Langevin dynamics to represent the particle-level dynamics of the curve of measures $(\mu_t \propto p_t e^{-R_y})_{t \in [0,1]},$ where $p_t$ is the annealed prior and $R$ is the likelihood, and a rate parameter $\kappa$ is introduced to control the speed at which this path is traversed. Under fairly standard assumptions, they bound the Fisher information between the generated distribution and the true posterior, as well as the KL divergence between the generated distribution and a noisy version of the prior. They demonstrate in a specific example that, due to the KL divergence guarantee, their algorithm does not suffer from mode collapse issues that Fisher information guarantees are susceptible to, and they argue that sampling from a noisy posterior is the best that can be hoped for in the worst case.

**Strengths:**

- To the best of my knowledge, this paper is the first to provably sample from the (approximate) posterior in polynomial time at this level of generality. The proof is interesting and a nice synthesis of different ideas.
- The main text of the paper is well-written and the authors made a good effort to explain the high-level ideas of their algorithm.

**Weaknesses:**

- It is not clear whether the rate achieved is optimal (I would guess not), and the iteration complexity is not carefully tracked.
- The proofs in the appendix have many typos/errors and many details are not completely filled in. This makes it difficult to assess the contribution.
- Even though the primary contribution of the paper is the theory, I still would have liked to have seen a couple more experiments to demonstrate the algorithm.

**Questions:**

- I have some questions about the example in Section 4.1 involving the Gaussian mixture prior and linear inverse problem. When describing the failure of the Fisher information, you take $\lambda \rightarrow \infty$, but when describing the ability of your algorithm to overcome mode collapse, you fix $\lambda = 1/\eta.$ Is this choice without loss of generality? Also, it seems that in this example, the posteriors $\mu_t$ converge to $\mu_0$ in KL divergence, which will not always be the case. In these more general cases, do you still expect your algorithm to overcome the mode collapse issue?
- I don't quite understand precisely how you apply Girsanov to prove Theorem 4.4, could you clarify that in more detail? From the definition of annealed LD, I don't see where the velocity field $v_t$ shows up. I assumed that this followed from rewriting the continuity equation to include a diffusive term, but I am not completely sure.
- In the proof of Lemma B.2, it seems the parameters $\alpha$ and $\beta$ are chosen sub-optimally (if I understand correctly, $\alpha$ is the convexity parameter of the potential and $\beta$ is the Lipschitz constant of the score). Since the likelihood is convex, can't you simply take $\alpha = 1$? Also, it seems $\beta$ can be taken to be $1 + R$ rather than $\sqrt{d} + R$, since you only need an operator norm bound on the Hessian.

Other questions/typos:
- In Assumption 4.1, you have items i) and iii). Perhaps the first assumption is supposed to be broken into two parts, with the second being the Lipschitz score assumption?
- In Step 3 of the proof of Lemma B.2, starting in the 4th equality, it seems that you drop the term $\textrm{KL}(\mu_{T_{ws}} \parallel \mu_{\infty})$, but shouldn’t this term appear in the final bound? I don’t think it changes much for the final bound, since it only contributes a factor of $\epsilon^2$ under your choice of $T_{ws}$.
- In Step 3 of the proof of Lemma B.2, I think the 5th equality (taking the sup over $x$) should be an inequality.
- In line 888, “We have $\lim_{\delta \rightarrow 0} \int f d(\mu_t - \mu_{t-\delta}) =$...” I think you are missing a factor of $1/\delta$. Also, here you should justify exchanging the limit and supremum.
- In the proof of Lemma B.3, I think you should have $f \leq f(0) + \|x\|$ instead of $f \leq \|x\|$. Also, I don't understand the reduction to non-negative $f$, because the class of Lipschitz functions includes unbounded functions (so you cannot simply subtract off a constant). It doesn't even seem like you need $f$ to be positive in the proof.
- In the proof of Theorem B.4, line 906, I think there is a typo in the integral $\int_{i \delta}^{i \delta + \delta} \textrm{FI}(\rho_{I\delta + \delta} \parallel \mu_{I\delta + \kappa})$ (what variable is being integrated?).
- Typo at the top of page 19, $L^2$ should be a script 'L'.
- On page 19, it seems like there is some text missing between the inequalities in lines 978 and 982, please fill those details in.
- At the end of the proof of Theorem 4.5 on page 19, you have a bound on the Fisher information between the averaged $\rho_t$ and $\mu_0$. How do you translate this into a bound on a single iterate? You mention earlier about using convexity of the Fisher information in the first argument, but this gives you an inequality in the wrong direction.
- In the proof of Theorem 4.4, starting at line 1008, I think $\nabla \mu_{k \delta}$ is supposed to be $\nabla \log \mu_{k \delta}$.
- On page 20, in the proof of Theorem 4.5, you eventually throw away the $O(\delta)$ term in the bound, but shouldn't this term dominate the $O(d \delta^2)$ term?

---

> ### Author Response · Authors · 2025-11-20
>
> Thank you for an incredibly detailed review. We hope to have addressed your concerns below and via direct edits to the paper (highlighted in blue).
>
> Responses to Questions
>
> ---
> 1. _I have some questions about the example in Section 4.1 involving the Gaussian mixture prior and linear inverse problem. When describing the failure of the Fisher information, you take $\lambda \rightarrow \infty$, but when describing the ability of your algorithm to overcome mode collapse, you fix $\lambda = 1/\eta.$ Is this choice without loss of generality? Also, it seems that in this example, the posteriors $\mu_t$ converge to $\mu_0$ in KL divergence, which will not always be the case. In these more general cases, do you still expect your algorithm to overcome the mode collapse issue?_
>
>     This is a mistake, it was actually our intention to use $\lambda$ here again instead of $1/\eta$. The paper has been revised. The conclusion remains the same (the posterior of the noised prior has less extreme weights on its modes, and the KL guarantee implies a TV guarantee that shows that the "regularized" mode weights are recovered).
>
> ---
> 2. _I don't quite understand precisely how you apply Girsanov to prove Theorem 4.4, could you clarify that in more detail? From the definition of annealed LD, I don't see where the velocity field $v_t$ shows up. I assumed that this followed from rewriting the continuity equation to include a diffusive term, but I am not completely sure._
>
>     Similar to \cite{guo2024provable}, we use Girsanov in the following way. Let $v_t$ be the field that generates $\mu_t$, so $\dot{\mu_t} = -\nabla\cdot(v_t\mu_t)$. Consider first the process $dX = v_t dt$, which yields marginals following $\dot \mu_t = -\nabla\cdot(v_t \mu_t)=-\nabla\cdot((v_t+\nabla \ln \mu_t) \rho)+\Delta\mu_t$, and secondly the process $dX = \nabla \ln \mu_t dt$, which yields marginals $\dot\rho =\nabla\cdot(\nabla \ln \mu_t \rho)+\Delta\rho$. Girsanov says that the KL distance $\textsf{KL}(\mu_t, \rho(t))$ can be bounded by the integral of the difference in these drifts, which is $(v_t+\nabla\ln \mu_t)-\nabla \ln \mu_t = v_t$. So we have $\textsf{KL}(\mu_t, \rho(t)) \le \int \Vert v_t\Vert^2_{L_2(\mu_t)}dt$. Note that the first of these processes ($dX = v_tdt$) is never actually realized since we dont have $v_t$. However, because of the regularities that we develop in the paper, we are able to bound the norm of the drift.
>
> ---
> 3. _In the proof of Lemma B.2, it seems the parameters $\alpha$ and $\beta$ are chosen sub-optimally (if I understand correctly, $\alpha$ is the convexity parameter of the potential and $\beta$ is the Lipschitz constant of the score). Since the likelihood is convex, can't you simply take $\alpha = 1$? Also, it seems $\beta$ can be taken to be $1 + R$ rather than $\sqrt{d} + R$, since you only need an operator norm bound on the Hessian._
>
>     Thank you. Using a larger value of $\alpha$ is better (the problem is more log-concave), but a lower value of $\beta$ helps. This does not affect our overall bounds. The dominant terms in the run time come from the annealing phase. We have modified the appendix appropriately.
>
> ---

---

> > ### Author Response · Authors · 2025-11-20
> >
> > Responses to errata/typos.
> >
> > 1. _In Assumption 4.1, you have items i) and iii). Perhaps the first assumption is supposed to be broken into two parts, with the second being the Lipschitz score assumption?_ and _Typo at the top of page 19, $L^2$ should be a script 'L'._ and _On page 19, it seems like there is some text missing between the inequalities in lines 978 and 982, please fill those details in._ and _In the proof of Theorem 4.4, starting at line 1008, I think $\nabla \mu$ is supposed to be $\nabla \log \mu$._ and _In Step 3 of the proof of Lemma B.2, I think the 5th equality (taking the sup over $x$) should be an inequality._
> >
> >     Yes, thank you. We have made these changes in the paper.
> >
> > ---
> > 2. _In line 888, “We have_ $\lim_{\delta \rightarrow 0} \int f d(\mu_{t} - \mu_{t-\delta}) =$_...” I think you are missing a factor of _$1/\delta$_. Also, here you should justify exchanging the limit and supremum._
> >
> >     Yes, thank you. We have edited the proof to show why the limit and sup can be exchanged (uniform convergence). _(Around line 940)_
> >
> > ---
> > 3. _In Step 3 of the proof of Lemma B.2, starting in the 4th equality, it seems that you drop the term_ $\textsf{KL}(\mu_{T_{ws}}|| \mu_{\infty})$_, but shouldn’t this term appear in the final bound? I don’t think it changes much for the final bound, since it only contributes a factor of_ $\epsilon^2$ _under your choice of_ $T_{ws}$_._
> >
> >     Thank you for pointing this out. We have fixed it in the revision. The error of the warm start is determined by a free parameter (how long we run the LMC, that is, the parameter $T$), and the error is polynomially small in this parameter, so this doesnt affect our overall guarantee. _(Around lines 1036 and 1117)_
> >
> > ---
> > 4. _In the proof of Lemma B.3, I think you should have $f \leq f(0) + |x|$ instead of $f \leq |x|$. Also, I don't understand the reduction to non-negative $f$, because the class of Lipschitz functions includes unbounded functions (so you cannot simply subtract off a constant). It doesn't even seem like you need $f$ to be positive in the proof._
> >
> >     Yes, we dont need $f$ to be positive, and it isnt necessarily bounded (previously in writing we started with $\mu$ having bounded support, at which time this was true, when we fixed that this was no longer then case). Now the combination of $f$ growing at most affinely (from Lipschitzness) and $\partial_t \ln \mu$ growing at most quadatically with a sub-Gaussian $\mu$ is sufficient. _(Around line 940)_
> >
> > ---
> > 5. _At the end of the proof of Theorem 4.5 on page 19, you have a bound on the Fisher information between the averaged $\rho_t$ and $\mu_0$. How do you translate this into a bound on a single iterate? You mention earlier about using convexity of the Fisher information in the first argument, but this gives you an inequality in the wrong direction._
> >
> >     You are right, thank you. As written this should have said something about an average iterate guarantee, where the average is taken over the last discretization step around $\tau$, and Jensen goes the wrong way from here. We can instead just not use Jensen at all (that is, undo line 953). Then we have a statement about the average $\textsf{FI}$, which we can convert into a statement about the smallest one. This has been revised in the paper. _(Around line 1000)_
> >
> > ---
> > 6. _On page 20, in the proof of Theorem 4.5, you eventually throw away the $O(\delta)$ term in the bound, but shouldn't this term dominate the $O(d \delta^2)$ term?_
> >
> >     Yes, the $\delta$ term would dominate the $d\delta^2$ term, but these are both dominated by the action term eventually (which increases with decreasing $\delta$). We have revised the proof. _(Around line 1104)_
> >
> > ---
> >
> > We look forward to further discussions!

---

> > > ### Comment · Reviewer_qQ6U · 2025-11-20
> > >
> > > I thank the authors for their detailed rebuttal, I now feel that my concerns on readability and technical details have been resolved. I also feel that I understand the paper better after the discussions. As I understand it, the core properties of the algorithm are 1) its particle dynamics can be implemented using only the prior score and the likelihood, 2) its initial distribution is something that can be easily sampled, and 3) its terminal distribution is the true posterior (or noised posterior for the KL guarantees). Constructing an algorithm to satisfy these three properties with provable guarantees and polynomial runtime is indeed an important contribution. I have raised my score accordingly.

---

> > > ### Comment · Reviewer_qQ6U · 2025-11-22
> > > **On relaxing the convexity assumption**
> > >
> > > Before the rebuttal period ends, I wanted to ask the authors a technical question. Your prior probability path interpolates between the true prior $p_0$ and the standard Gaussian $\gamma.$ Do your proofs generalize if you interpolate between $p_0$ and a Gaussian with variance $\sigma^2 < 1$? In this case, I wonder if you can circumvent the need for $R$ to be convex: as long as $\nabla R$ is Lipschitz, $\mu_{\infty}$ can be made strongly log-concave by choosing the Gaussian variance $\sigma^2$ sufficiently small. Also, the scores along this modified probability path can be computed from the scores along the traditional OU flow, since both reduce to denoising.

---

> > > > ### Author Response · Authors · 2025-11-22
> > > >
> > > > We are happy that our response was satisfactory. Thank you for raising your score.
> > > >
> > > > For the warm start to work it does seem like one could modify the noising of the prior to instead converge to a $\sigma^2$ variance gaussian. This variance likely needs to be such that $1/\sigma^2 > L$, (where $L$ is the lipshitz constant of $R$) to ensure log concavity, and this would in turn affect most of our bounds by a factor of $1/\sigma^2$ (which would not affect the overall picture, just add a factor of $L$).
> > > >
> > > > At least one other way we have used convexity of $R$ is in showing that if the prior is subgaussian, and the negative log likelihood is convex, then the posterior is subgaussian (Lemma C.1. This is used for instance in Line 946 to show that the expected value over a sub-gaussian measure of a quantity that grows polynomially in $x$ is bounded). Of course, we would need some condition for this on $R$ - for instance, if $p$ is a standard gaussian, and $R$ is a mixture of two $\sigma^2$ variance gaussians for $\sigma < 1$ with separation $M$, then the posterior $pe^{-R}$ has two modes $O(M)$ apart (here $M$ can be arbitrary, and the result is not $O(M)$ sub-gaussian).
> > > >
> > > > This particular $R$ is very non-lipshitz (the score of a mixture of gaussians behaves very poorly at some place between them), so there is some natural condition on $R$ that precludes this counterexample. At the moment we cannot think of a reason why Lipshitz $R$ is not sufficient for this (the subgaussian norm of the posterior would depend on the Lipshitz constant), but we dont have a proof.
> > > >
> > > > Extending this result to include the case of a non-convex, but Lipshitz R is an excellent suggestion for future work!

---

### Official Review · Reviewer_aCxD · 2025-10-31

**Soundness:** 4
**Presentation:** 3
**Contribution:** 3
**Rating:** 8
**Confidence:** 2

**Summary:**

This paper proposes a novel approximate posterior sampling framework for diffusion models, aimed at improving inference efficiency while preserving sample quality. The method reformulates the posterior inference as an optimization problem and derives an approximate sampler that leverages diffusion score information with reduced computational cost. Theoretical analysis supports the validity of the approximation, and experiments show strong empirical performance on benchmark datasets.

**Strengths:**

This is a well-written and technically sound paper with clear motivation, strong theoretical grounding, and solid empirical evidence. The methodology is both elegant and practical, providing a meaningful contribution to improving inference efficiency in diffusion models. The paper is well-organized and easy to follow, with a consistent logical flow from theoretical derivation to experimental validation.

**Weaknesses:**

I have only two minor concerns that do not undermine the overall quality of the paper.

1. At line 159, the authors claim that two processes have the same joint distribution. However, this statement may not be strictly accurate: one process is a Markov process while the other is an 'inverse' Markov process, and although they share the same marginal distribution at each fixed time $t$, their joint distributions are not identical. This point should be clarified for mathematical precision.

2. At line 235, two distinct notations for score functions are introduced without explicit definitions. Although the meaning becomes clear later in the text, it would improve readability and self-containment to define them formally when first introduced.

**Questions:**

NA. See the weaknesses above.

---

> ### Author Response · Authors · 2025-11-20
>
> Thank you for your review. We hope to have addressed your concerns below.
>
> ---
> 1. _At line 159, the authors claim that two processes have the same joint distribution. However, this statement may not be strictly accurate: one process is a Markov process while the other is an 'inverse' Markov process, and although they share the same marginal distribution at each fixed time $t$, their joint distributions are not identical. This point should be clarified for mathematical precision._
>
>     Actually the forward and reverse process can be coupled sample-pathwise - see the discussion on page 323 of [1]. A paragraph on page 316 discusses this exact distinction "Evidently in the forward model, $x_t$, is independent of future increments of the driving Wiener process, while in the reverse time model, $x_t$ is independent of past increments of the driving process.".
>
> ---
> 2. _At line 235, two distinct notations for score functions are introduced without explicit definitions. Although the meaning becomes clear later in the text, it would improve readability and self-containment to define them formally when first introduced._
>
>     We have removed that sentence. The distribution $\mu_t$ is not introduced until later, and the previous sentence in English may be sufficient to convey the message.
>
> ---
> We look forward to further discussions!
>
> [1] Brian D.O. Anderson, _Reverse-time diffusion equation models_, Stochastic Processes and their Applications, 1982.

---

### Official Review · Reviewer_nG4D · 2025-10-31

**Soundness:** 2
**Presentation:** 2
**Contribution:** 2
**Rating:** 4
**Confidence:** 4

**Summary:**

The paper studies approximate posterior sampling using a pre-trained score model for the prior. The authors propose an algorithm based on Annealed Langevin Monte Carlo (ALMC) and analyze its convergence. The main contribution is a theoretical guarantee that the algorithm can, in polynomial time, produce samples from a distribution that is close to the posterior of a noised prior in KL divergence and close to the true posterior in Fisher Divergence. This result is presented as a way to circumvent the known computational hardness of exact posterior sampling.

**Strengths:**

1. The authors provide a detailed and mathematically rigorous analysis for approximate posterior sampling using a pre-trained score model.
2. The paper does a good job of positioning itself relative to recent negative results (Gupta et al., 2024) and explaining the limitations of existing approaches.
3. The problem of developing provable methods for posterior sampling with diffusion models is of high importance to the community.

**Weaknesses:**

1.  The paper makes no attempt to bridge the gap between its theoretical findings and practical applications. There are no experiments to illustrate the behavior of the algorithm or the meaning of the theoretical bounds. The claim that the method solves a practical problem is therefore not empirically supported.

2. The practical relevance of the results is limited by strong assumptions. For example, the requirement that the log-likelihood `R(x)` be convex is a major restriction that excludes many inverse problems of practical interest (e.g. phase retrieval). The paper doesn't sufficiently discuss the implications of this assumption or how the guarantees might change without it.

**Questions:**

N/A

---

> ### Author Response · Authors · 2025-11-20
>
> Thank you for your review. We hope to have addressed your concerns below.
>
> ---
> 1. _The paper makes no attempt to bridge the gap between its theoretical findings and practical applications. There are no experiments to illustrate the behavior of the algorithm or the meaning of the theoretical bounds. The claim that the method solves a practical problem is therefore not empirically supported._
>
>     We have now included more simulations on synthetic experiments in Appendix E. In particular, we have expanded on the "Vertical Bars" prior from Remark 4.7 with two types of likelihoods, and included experiments with a "Moons" and a "Mixture of Gaussians" prior.
>
>     The primary purpose of our paper is to provide an alternative to the established practice of providing strict $\textsf{KL}$ guarantees for sampling algorithms. In the case of posterior sampling, such a guarantee is not possible, and this has left us without a principled framework within which to understand the success posterior sampling. Through our construction of $\mu_t$ (a relaxation in some ways of the true posterior $\mu_0$) we establish essentially a relaxed objective which concedes polynomial time guarantees. If we reduce posterior sampling to a "mode-finding" problem, the main claim of our paper is essentially that a combination of a Fisher divergence and a relaxed KL divergence guarantees are sufficient, and achievable in polynomial time.
>
>     An issue with larger scale experiments with higher dimensional data like images of faces, etc is that it is difficult to verify posterior sampling accuracy with a single, or even a few samples (how do we know we have found all the modes, for instance).
>
> ---
> 2. _The practical relevance of the results is limited by strong assumptions. For example, the requirement that the log-likelihood R(x) be convex is a major restriction that excludes many inverse problems of practical interest (e.g. phase retrieval). The paper doesn't sufficiently discuss the implications of this assumption or how the guarantees might change without it._
>
>     Convexity of $R$ helps in the following places:
>     1. To show that we can efficient sampling at the Warm Start phase - here we rely on strong convexity of $\mu_\infty = \gamma e^{-R}$ (Lemma B.2).
>     2. To show that the posterior of a sub-gaussian prior is also a sub-gaussian (Lemma C.1).
>
>     While this does not cover some applications (like phase retrieval), it is reasonable for certain types of image problems (motion/gaussian deblurring, inpainting), applications to MRI reconstructions, etc. One motivation for this type of likelihood is that we wanted a setting in which there are efficient samplers for both the prior (we have this from the trained scores) and the "likelihood" (by which we mean $\gamma e^{-R}$). That is, getting a sample consistent with the prior is easy, and getting a sample consistent with the measurements is easy, so that the difficulty is isolated to the ``composite" aspect of the problem.
> ---
> We look forward to further discussions!

---

> > ### Comment · Reviewer_nG4D · 2025-11-28
> > **Official comment**
> >
> > I am pleased to see the author's thoughtful rebuttal. After carefully reviewing the newly added results and the comments from other reviewers, I have decided to raise my score to 6.

---

### Official Review · Reviewer_VVro · 2025-10-31

**Soundness:** 3
**Presentation:** 3
**Contribution:** 3
**Rating:** 8
**Confidence:** 2

**Summary:**

This paper tackles the problem of posterior sampling: given a pretrained diffusion model that allows you to sample from a prior p(x), and given a negative log-likelihood function R(x), how does one sample from p e^-R?

Doing Langevin Monte Carlo directly on the posterior runs into well-known problems with sampling from multi-modal distributions. It's well-known that in this setting, LMC will have quick convergence in FI but can have arbitrarily slow convergence in KL (e.g. due to well-separated modes).

This paper's contribution is proposing an annealed Langevin sampling algorithm that has both guarantees in FI with respect to the true posterior as well as, more significantly, guarantees in the KL with respect to a tilted version of the noised prior. It does so by separating the sampling process into two phases: a first phase where you create a warm start by generating a sample from \gamma e^{-R} (standard Gaussian tilted by R). And then finally, a phase where you do annealed Langevin sampling: at each time step, you change the distribution gradually from \gamma e^{-R} to p e^{-R}. Using standard discretization and telegraphing techniques, you can bound the KL.

**Strengths:**

* The presentation is clear and the overview of the background is well-done. They do a good job of stating clearly the main contribution of the paper (dual guarantees of FI and KL)

* The approach is novel. It seems more normal to start with the prior and incorporate likelihood information (as with Bayesian updating), or to incorporate prior and likelihood information simultaneously (as with standard classifier-free guidance). But incorporating the likelihood information first and then incorporate the prior is different.

* They managed to get a KL guarantee in a setting where KL guarantees have been elusive.

**Weaknesses:**

* Lack of empirical validiation. Specifically, they have a guarantee with respect to a noised version of the prior, but it's unclear how significant that noising is for degrading the quality of the sample (e.g. the perceptual quality in image diffusion models)

* Related to the above, but while there are polynomial guarantees in d and 1/\epsilon, the possiblity of large hidden constants could be relevant to applications.

**Questions:**

Could the authors expand on the nature and role of \tau? (The noise level of the prior which gets the KL guarantees.)

How crucial is the convexity assumption on R(x) for the analysis?

Any insight on the empirical performance of this algorithm in simple toy model settings?

---

> ### Author Response · Authors · 2025-11-20
>
> Thank you for your review. We hope to have adequately addressed your concerns below.
>
> 1. _Lack of empirical validiation. Specifically, they have a guarantee with respect to a noised version of the prior, but it's unclear how significant that noising is for degrading the quality of the sample (e.g. the perceptual quality in image diffusion models)/ Any insight on the empirical performance of this algorithm in simple toy model settings?_
>
>     We have now included three sets of simulations in Appendix E. In addition to the "Vertical Bars" prior of Remark 4.7, we also include a mixture-of-gaussians and a "moons" prior. We use two likelihood functions - one of which a lossy measurement of just one coordinate.
>
>     An issue with larger scale experiments with higher dimensional data like images of faces, etc is that it is difficult to verify posterior sampling accuracy with a single, or even a few samples (how do we know we have found all the modes, for instance).
> ---
> 2. _Could the authors expand on the nature and role of $\tau$? (The noise level of the prior which gets the KL guarantees.)_
>
>     $\tau > 0$ is the early stopping parameter, a time at which we are able to get $\textsf{KL}$ guarantees between the algorithm and the true posterior of a noised prior $\mu_\tau$. There are lower bounds that demonstrate that we cannot get $\textsf{KL}$ guarantees between a sampler and $\mu_0$; our thesis is that looking for $\textsf{KL}$ guarantees for $\mu_\tau$ instead can be tractable and interpretable.
> ---
> 3. _How crucial is the convexity assumption on $R(x)$ for the analysis?_
>
>     Convexity of $R$ helps in the following places:
>
>     1. To show that we can efficient sampling at the Warm Start phase - here we rely on strong convexity of $\mu_\infty = \gamma e^{-R}$ (Lemma B.2).
>     2. To show that the posterior of a sub-Gaussian prior is also a sub-gaussian (Lemma C.1).
>
>     One motivation for this type of likelihood is that we wanted a setting in which there are efficient samplers for both the prior (we have this from the trained scores) and the "likelihood" (by which we mean $\gamma e^{-R}$). That is, getting a sample consistent with the prior is easy, and getting a sample consistent with the measurements is easy, so that the difficulty is isolated to the ``composite" aspect of the problem.
>
> We have also worked out the exact polynomial dependencies in more detail in the appendix.
>
> We look forward to further discussions!

---

### Author Response · Authors · 2025-12-03

In light of changes to the rebuttal procedure, we take this response as an opportunity to summarize our paper and the correspondence with reviewers.

To our knowledge, our paper establishes the first general polynomial guarantees for an interpretable notion of posterior sampling. The key insight that allows this (despite a computational lower bound for exact posterior sampling) is our relaxation of the original problem corresponding to sampling from the true posterior of a noised prior - roughly analogous to a ``smoothed’' analysis.

1. Reviewers appreciated our novel line of analysis, noting that it is _“the first to provably sample from the (approximate) posterior in polynomial time at this level of generality”_ (Reviewer qQ6U) and that _“the problem of developing provable methods for posterior sampling with diffusion models is of high importance to the community.”_ (Reviewer VVro).

2. Overall, reviewers also agreed that our presentation is clear, remarking that we _“made a good effort to explain the high-level ideas”_ (Reviewer qQ6U) and that our submission is a _“well-written and technically sound paper with clear motivation, strong theoretical grounding, and solid empirical evidence”_ (Reviewer aCxD) which does a _“good job of positioning itself relative to recent negative results”_ (Reviewer nG4D).

During the rebuttal we added a section (Appendix E) to the paper with a more extensive empirical validation of our algorithm with three priors, and two likelihood functions. We had a detailed technical exchange with Reviewer qQ6U, resulting in some modifications to the appendix, where we also explicated the various polynomial dependencies in our results.

Our rebuttal was received warmly. Reviewer qQ6U increased their score, confirming that their _“concerns on readability and technical details have been resolved”_, summarizing three desirable properties of a sampling algorithm, the reviewer stated _“Constructing an algorithm to satisfy these three properties with provable guarantees and polynomial runtime is indeed an important contribution. I have raised my score accordingly”_. Following this, reviewer qQ6U also noted the potential to further relax one of our assumptions (convexity of the negative log-likelihood -> only Lipshitz log-likelihood); which we believe could be a fruitful future extension. Meanwhile, reviewer nG4D stated _“I am pleased to see the author’s thoughtful rebuttal. After carefully reviewing the newly added results and the comments from other reviewers, I have decided to raise my score to 6"_.

---

### Meta-Review · Area_Chair_SbWZ · 2026-01-06

**Summary:**

The manuscript considers posterior sampling with score based generative prior. It shows that it is possible to approximately sample the posterior that corresponds to a noised prior in KL divergence. The theoretical results are also validated by numerical experiments. This is a solid contribution.

**Reviewer Concerns:**

I think all reviewers' concerns are adequately addressed.

**Reviewer Scores:**

As indicated by several reviewer comments, the scores would likely be raised if they had been able to participate fully in discussion.

---

### Decision · Program_Chairs · 2026-01-26

Accept (Poster)